# COMPOSITIONAL LANGUAGES EMERGE IN A NEURAL ITERATED LEARNING MODEL

**Yi Ren,[1] Shangmin Guo,[2] Matthieu Labeau,[3] Shay B. Cohen,[1] Simon Kirby[1]**
[1] University of Edinburgh, United Kingdom, [2] University of Cambridge, United Kingdom
[3] LTCI, Télécom Paris, Institut Polytechnique de Paris, France
[1] `renyi.joshua@gmail.com, scohen@inf.ed.ac.uk, simon.kirby@ed.ac.uk`
[2] `sg955@cam.ac.uk,` [3] `matthieu.labeau@telecom-paris.fr`

## ABSTRACT

The principle of compositionality, which enables natural language to represent complex concepts via a structured combination of simpler ones, allows us to convey an open-ended set of messages using a limited vocabulary. If compositionality is indeed a natural property of language, we may expect it to appear in communication protocols that are created by neural agents in language games. In this paper, we propose an effective neural iterated learning (NIL) algorithm that, when applied to interacting neural agents, facilitates the emergence of a more structured type of language. Indeed, these languages provide learning speed advantages to neural agents during training, which can be incrementally amplified via NIL. We provide a probabilistic model of NIL and an explanation of why the advantage of compositional language exist. Our experiments confirm our analysis, and also demonstrate that the emerged languages largely improve the generalizing power of the neural agent communication.

## 1 INTRODUCTION

Natural language understanding (NLU), which is exemplified by challenging problems such as machine reading comprehension, question answering and machine translation, plays a crucial role in artificial intelligence systems. So far, most of the existing methods focus on building statistical associations between textual inputs and semantic representations, e.g. using first-order logic (Manning et al., 1999) or other types of representations such as abstract meaning representation (Banarescu et al., 2013). Recently, *grounded language learning* has gradually attracted attention in various domains, inspired by the hypothesis that early language learning was focused on problem-solving (Kirby & Hurford, 2002). While related to NLU, it focuses on the *pragmatics* (Clark, 1996) of learning natural language, as it implies learning language from scratch, grounded in experience. This research is often practiced through the development of neural agents which are made to communicate with each other to accomplish specific tasks (for example, playing a game). During this process, the agents build mappings between the concepts they wish to communicate about and the symbols used to represent them. These mappings are usually referred to as 'emergent language'.

So far, an array of recent work (Havrylov & Titov, 2017; Mordatch & Abbeel, 2018; Kottur et al., 2017; Foerster et al., 2016) has shown that in many game settings, the neural agents can use their emergent language to exchange useful coordinating information. While the best way to design games to favour language emergence is still open to debate, there is a consensus on the fact that we should gear these emergent languages towards sharing similarities with natural language. Among the properties of natural language, *compositionality* is considered to be critical, because it enables representation of complex concepts through the combinination of several simple ones. While work on incorporating compositionality into emergent languages is still in its early stage, several experiments have already demonstrated that by properly choosing the maximum message length and vocabulary size, the agents can be brought together to develop a compositional language that shares similarities with natural language (Li & Bowling, 2019; Lazaridou et al., 2018; Cogswell et al., 2019).

In a different body of language research literature, evolutionary linguists have already studied the origins of compositionality for decades (Kirby & Hurford, 2002; Kirby et al., 2014; 2015). They

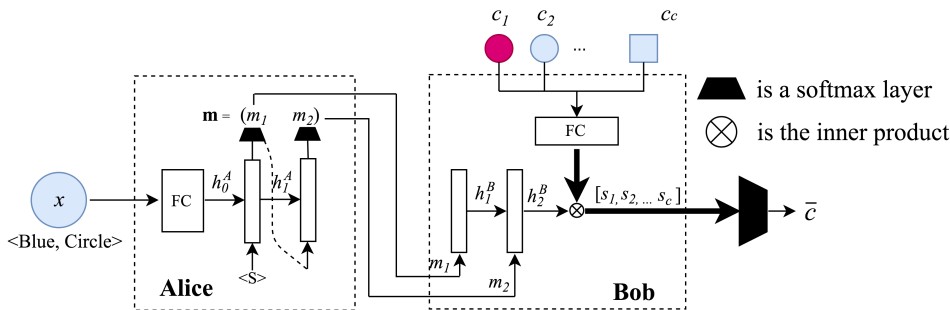

Figure 1: Referential communication game and architectures of the agents.

proposed a cultural evolutionary account of the origins of compositionality and designed a framework called iterated learning to simulate the language evolution process, based on the idea that the simulated language must be learned by new speakers at each generation, while also being used for communication. Their experiments show that highly compositional languages may indeed emerge through iterated learning. However, the models they introduced were mainly studied by means of experiments with human participants, in which the compositional languages is favored by the participants because human brain favors structure. Hence, directly applying this framework to ground language learning is not straightforward: we should first verify the existence of the preference of compositional language at the neural agent, and then design an effective training procedure for the neural agent to amplify such an advantage.

In this project, we analyze whether and how the learning speed advantage of the highly compositional languages exists in the context of communication between neural agents playing a game. Then we propose a three-phase neural iterated learning algorithm (NIL) and a probabilistic explanation of it. The experimental results demonstrate that our algorithm can significantly enhance the topological similarity (Brighton & Kirby, 2006) between the emergent language and the original meaning space in a simple referential game (Lewis, 1969). Such highly compositional languages also generalize better, because they perform well on a held-out validation set. We highlight our contribution as:

- We discover the learning speed advantages of languages with *high topological similarity* for neural agents communicating in order to play a referential game.

- We propose the NIL based on those advantages, which is quite robust compared to most of the related works.

- We propose a probabilistic framework to explain the mechanisms of NIL.

## 2 BACKGROUND

### 2.1 REFERENTIAL GAME

We analyze a typical and straightforward object selection game, in which a speaking agent (Alice, or speaker) and a listening agent (Bob, or listener) must cooperate to accomplish a task. In each round of the game, we show Alice a target object $x$ selected from an object space $\mathcal{X}$ and let her send a discrete-sequence message $\mathbf{m}$ to Bob. We then show Bob $c$ different objects ($x$ must be one of them) and use $c_1, ..., c_c \in \mathcal{X}$ to represent these candidates. Bob must use the message received from Alice to select the object that Alice refers among the $c$ candidates. If Bob's selection $\bar{c}$ is correct, both Alice and Bob are rewarded. The objects are shuffled and candidates are randomly selected in each round to avoid the agents recognizing the objects using their order of presentation.

In our game, each object in $\mathcal{X}$ has $N_a$ attributes (color and shape are often used in the literature), and each attribute has $N_v$ possible values. To represent objects, similarly to the settings chosen in (Kottur et al., 2017), we encode each attribute as a one-hot vector and concatenate the $N_a$ one-hot vectors to represent one object. The message delivered by Alice is a fixed-length discrete sequence $\mathbf{m} = (m_1, ..., m_{N_L})$, in which each $m_i$ is selected from a fixed size meaningless vocabulary $V$.

## 2.2 Neural Agent Structures

Neural agents usually have separate modules for speaking and listening, which we name Alice and Bob. Their architectures, shown in Figure 1, are similar to those studied in (Havrylov & Titov, 2017) and (Lazaridou et al., 2018). Alice first applies a multi-layer perceptron (MLP) to encode $x$ into an embedding, then feeds it to an encoding LSTM (Hochreiter & Schmidhuber, 1997). Its output will go through a softmax layer, which we use to generate the message $m_1, m_2, \cdots$. Bob uses a decoding LSTM to read the message and uses a MLP to encode $c_1, ..., c_c$ into embeddings. Bob then takes the dot product between the hidden states of the decoding LSTM and the embeddings to generate scores $s_c$ for each object. These scores are then used to calculate the cross-entropy loss when training Bob. When Alice and Bob are trained using reinforcement learning, we can use $p_A(\mathbf{m}|x; \theta_A)$ and $p_B(\bar{c}|\mathbf{m}, c_1, ..., c_c; \theta_B)$ to represent their respective policies, where $\theta_A$ and $\theta_B$ contain the parameters of each of the neural agents. When the agents are trained to play the game together, we use the REINFORCE algorithm (Williams, 1992) to maximize the expected reward under their policies, and add the entropy regularization term to encourage exploration during training, as explained in (Mnih et al., 2016). The gradients of the objective function $J(\theta_A, \theta_B)$ are:

$$\nabla_{\theta_A} J = \mathbb{E}\left[R(\bar{c}, x)\nabla \log p_A(\mathbf{m}|x)\right] + \lambda_A \nabla H[p_A(\mathbf{m}|x)] \tag{1}$$

$$\nabla_{\theta_B} J = \mathbb{E}\left[R(\bar{c}, x)\nabla \log p_B(\bar{c}|\mathbf{m}, c_1, ..., c_c)\right] + \lambda_B \nabla H[p_B(\bar{c}|\mathbf{m}, c_1, ..., c_c)], \tag{2}$$

where $R(\bar{c}, x) = \mathbb{1}(\bar{c}, x)$ is the reward function, $H$ is the standard entropy function, and $\lambda_A, \lambda_B > 0$ are hyperparameters. A formal definition of the agents can be found in Appendix C.

## 2.3 Measuring Compositionality

Compositionality is a crucial feature of natural languages, allowing us to use small building blocks (e.g., words, phrases) to generate more complex structures (e.g., sentences), with the meaning of the larger structure being determined by the meaning of its parts (Clark, 1996). However, there is no consensus on how to quantitatively assess it. Besides a subjective human evaluation, *topological similarity* has been proposed as a possible quantitative measure (Brighton & Kirby, 2006).

To define topological similarity, we first define the language studied in this work as $\mathcal{L}(\cdot) : \mathcal{X} \mapsto \mathcal{M}$. Then we need to measure the distances between pairs of objects: $\Delta_{\mathcal{X}}^{ij} = d_{\mathcal{X}}(x_i, x_j)$, where $d_{\mathcal{X}}(\cdot)$ is a distance in $\mathcal{X}$. Similarly, we compute the corresponding quantity for the associated messages $m_i = \mathcal{L}(x_i)$ in the message space $\mathcal{M}$ with $\Delta_{\mathcal{M}}^{ij} = d_{\mathcal{M}}(m_i, m_j)$, where $d_{\mathcal{M}}(\cdot)$ is a distance in $\mathcal{M}$. Then the topological similarity (i.e., $\rho$) is defined as the correlation between these quantities across $\mathcal{X}$. Following the setup of (Lazaridou et al., 2018) and (Li & Bowling, 2019), we use the negative cosine similarity in the object space and Levenshtein distances (Levenshtein, 1966) in the message space. We provide an example in Appendix B to give a better intuition about this metric.

## 3 Neural Iterated Learning Model

The idea of iterated learning requires the agent in current generation be partially exposed to the language used in the previous generation. Even this idea is proven to be effective when experimenting with human participants, directly applying it to games played by neural agents is not trivial: for example, we are not sure where to find the preference for high-$\rho$ languages for the neural agents. Besides, we must carefully design an algorithm that can simulate the "partially exposed" procedure, which is essential for the success of iterated learning.

### 3.1 Learning Speed Advantage for the Neural Agents

As mentioned before, the preference of high-$\rho$ language by the learning agents is essential for the success of iterated learning. In language evolution, highly compositional languages are favored because they are structurally simple and hence are easier to learn (Carr et al., 2017). We believe that a similar phenomenon applies to communication between neural agents:

**Hypothesis 1:** *High topological similarity improves the learning speed of the speaking neural agent.*

We speculate that high-$\rho$ languages are easier to emulate for a neural agent than low-$\rho$ languages. Concretely, that means that Alice, when pre-trained with object-message pairs describing a high-

$\rho$ language at a given generation, will be faster to successfully output the right message for each object. Intuitively, this is because the structured mapping described by a language with high $\rho$ is smoother, and hence has a lower sample complexity, which makes resulting examples easier to learn for the speaker agent (Vapnik, 2013).

**Hypothesis 2:** *High topological similarity allows the listening agent to successfully recognize more concepts, using less samples.*

We speculate that high-$\rho$ languages are easier for a neural agent to recognize. That means that Bob, when pre-trained with message-object pairs corresponding to a high-$\rho$ language, will be faster to successfully choose the right object. Intuitively, the lower topological similarity is, the more difficult it will be to infer unseen object-message pairs from seen examples. The more complex mapping of a low-$\rho$ language implies that more object-message pairs need to be provided to describe it. This translates as an inability for the listening agent to generalize the information it obtained from one object-message associated to a low-$\rho$ language to other examples. Thus, the general performance of Bob on any example will improve much faster when trained with pairs corresponding to a high-$\rho$ language than with a low-$\rho$ language. We provide experimental results in section 4.1 to verify our hypotheses. We also provide a detailed example in Appendix D to illustrate our reasoning.

### 3.2 NEURAL ITERATED LEARNING AND PROBABILISTIC ANALYSIS

We design the NIL algorithm to exploit these advantages in a robust manner, as detailed in Algorithm 1. The algorithm runs for $I$ generations: at the beginning of each generation $i$, both the agents are re-initialized. As Alice and Bob have different structures, they are then pre-trained differently (see line 5-7 for Alice and line 8-12 for Bob): this is the *learning phase*. Alice is pre-trained via categorical cross-entropy, using the data generated at the previous generation, which we denote $D_i$. Bob is pre-trained with REINFORCE, learning from the pre-trained Alice. We note $I_a$ and $I_b$ their respective number of pre-training iterations. With hypothesis 1, the expected $\rho$ of the language spoken by Alice should be higher than that of $D_i$. Meanwhile, Bob shold be more "familiar with" the language with a higher $\rho$ than $D_i$, as stated by hypothesis 2. Alice and Bob then play the game together for $I_g$ rounds in the *interacting phase*, in which both agents are updated via REINFORCE. In this phase, the languages used by them are filtered to be more unambiguous — their language must deliver information accurately to accomplish the task. Finally, in the *transmitting phase*, we feed all objects to Alice and let it output the corresponding messages to be stored in $D_{i+1}$ for the learning phase of the next generation.

To better understand how NIL enhances the expected $\rho$ of the languages generation by generation, we propose a probabilistic model for NIL in Appendix C, as well as a probabilistic analysis of the role played by Alice and Bob in every phase. Intuitively, at the beginning of each generation, the expected $\rho$ of language used by Alice (denoted by $\mathbb{E}_{\mathcal{L}}[\rho(\mathcal{L})]$) is quite low because of the random initialization. Then during the learning phase, Alice learns from $D_i$ and expected to have the same $\mathbb{E}_{\mathcal{L}}[\rho(\mathcal{L})]$ with $D_i$ if it perfectly learns that data set. However, as the high-$\rho$ language is favored by neural agent during training, the $\mathbb{E}_{\mathcal{L}}[\rho(\mathcal{L})]$ of the weakly pre-trained Alice should be higher than that of $D_i$. A similar thing may happen when pre-training Bob. Then in the interacting phase, as the game performance has no preference for language with different $\rho$, $\mathbb{E}_{\mathcal{L}}[\rho(\mathcal{L})]$ will not change in this phase.[1] Finally, in the transmitting phase, $D_{i+1}$ is sampled based on the language with current $\mathbb{E}_{\mathcal{L}}[\rho(\mathcal{L})]$, which is expected to be higher than that of $D_i$. In other words, $\mathbb{E}_{\mathcal{L}}[\rho(\mathcal{L})]$ would increase generation by generation (the details for derivations are provided in Appendix C):

$$\mathbb{E}_{\mathcal{L} \sim D_{i+1}}[\rho(\mathcal{L})] \geq \mathbb{E}_{\mathcal{L} \sim D_i}[\rho(\mathcal{L})]. \tag{3}$$

## 4 EXPERIMENTS AND DISCUSSIONS

In this section, we first verify hypotheses 1 and 2 by directly feeding languages with different $\rho$ to Alice and Bob. Then we examine the behavior and performance of the neural agents, as well as the expected $\rho$ of languages, at each generation. We conduct an ablation study, to examine the effect of pre-training Alice and Bob separately. We then investigate more thoroughly the advantages

---

[1] The role of interacting phase is to filter out those ambiguous language. This may change $\mathbb{E}_{\mathcal{L}}[\rho(\mathcal{L})]$, but without a preference for language with specific $\rho$.

---

Randomly initialize $D_1$
**for** $i = 1, 2, ..., I$ **do**
    Re-initialize Alice and Bob, get Alice$_i$ and Bob$_i$
    // ======= Learning Phase =======
    **for** $i_a = 1, 2, ..., I_a$ **do**
        Randomly sample an example pair from $D_i$ and use it to update Alice$_i$ with cross-entropy training
    **end for**
    **for** $i_b = 1, 2, ..., I_b$ **do**
        Alice$_i$ generates message based on input objects
        Bob$_i$ receives message and selects the target
        Bob$_i$ updates its parameters if rewarded
    **end for**
    // ======= Interacting Phase =======
    **for** $i_g = 1, 2, ..., I_g$ **do**
        Alice$_i$ generates message based on input objects
        Bob$_i$ receives message and selects the target
        **BOTH** Alice$_i$ and Bob$_i$ update parameters if rewarded
    **end for**
    // ======= Transmitting Phase =======
    **for** $i_s = 1, 2, ..., I_s$ **do**
        Generate object-message pairs by feeding objects to Alice$_i$ and save them to data set $D_{i+1}$
    **end for**
**end for**

**Algorithm 1:** The NIL algorithm. $I_a$, $I_b$ and $I_g$ are the number of iterations used to pre-train Alice, Bob, and to play the game at each generation.

---

brought by high-$\rho$ languages, and highlight the 'interval of advantage' in pre-training rounds, which could help in selecting reasonable $I_a$ and $I_b$. Finally, we conduct a series of experiments on a held-out validation set to highlight the positive effect of high-$\rho$ languages on the neural agents generalization ability — which shows the potential of iterated learning for NLU tasks. Details about our experimental setup and our choice of hyper-parameters can be found in Appendix A. More experiments about the robustness of NIL are presented in Appendix E.

## 4.1 LEARNING SPEED ADVANTAGES

We first use the experimental results in Figure 2 to verify hypotheses 1 and 2. In these experiments, we randomly initialize one Alice and feed languages with different expected $\rho$ for it to learn (and repeat the same procedure for Bob). We generate a perfect high-$\rho$ language ($\rho = 1$) using the method proposed in (Kirby et al., 2015), and randomly permute the messages to generate a low-$\rho$ language with $\rho = 0.21$. The other languages are intermediate languages generated during NIL. Note that there is no interacting nor transmitting phase in the experiment in this subsection: we only test the learning behavior of a randomly initialized Alice (or Bob) separately.

From the result in Figure 2-(a) and (b), we see that the high-$\rho$ languages indeed has the learning speed advantage at both the speaker and the listener side. One important finding is in Figure 2-(c), which record the expected $\rho$, i.e., $\mathbb{E}_{\mathcal{L}}[\rho(\mathcal{L})]$, during Alice's learning. From this figure, we find that when learning a language with low expected $\rho$, the value of $\mathbb{E}_{\mathcal{L}}[\rho(\mathcal{L})]$ will first increase, and finally converge to the $\rho$ of $D$. This phenomenon, caused by the learning speed advantage, makes the **weak pre-train** the essential design for the success of NIL: if $I_a$ is correctly chosen, we may expect a higher $\mathbb{E}_{\mathcal{L}}[\rho(\mathcal{L})]$ than that of the data set it learns from.

## 4.2 PERFORMANCE OF NIL

In this part, we record the game performance (i.e., the rate of successful object selections) and mean $\rho$ of the object-message pairs exchanged by the neural agents every 20 rounds. We run the simulation 10 times, with a different random number seed each time. Although the results are different, they

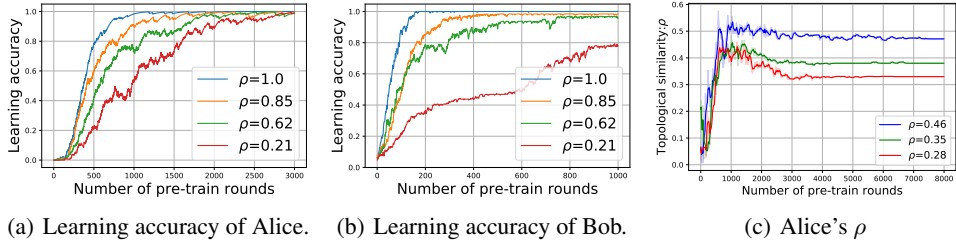

(a) Learning accuracy of Alice.    (b) Learning accuracy of Bob.    (c) Alice's $\rho$

Figure 2: Illustration of the learning speed of Alice and performance improving speed of Bob when pre-training is done with various languages of different topological similarities.

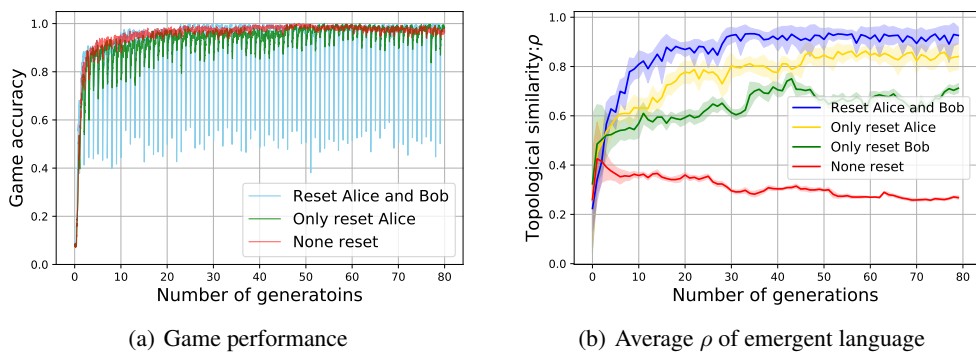

(a) Game performance          (b) Average $\rho$ of emergent language

Figure 3: Game performance and average topological similarity for the possible resetting strategies of our proposed iterated learning procedure of 80 generations. In these experiments, $I$=80 and $I_g$=4000, with all other hyper-parameters following Table 3.

all follow the same trend. In this first series of experiments, we compare the following 4 different methods:

- Iterated learning, with resetting both Alice and Bob at the beginning of each generation.
- Iterated learning, only resetting Alice at the beginning of each generation;
- Iterated learning, only resetting Bob at the beginning of each generation;
- No iterated learning: neither Alice nor Bob are reset during training.

From Figure 3-(a), we can see that for the 3 displayed variants of the procedure, neural agents can play the game almost perfectly after a few generations. The curve of the no-reset method will directly converge while the curves of the other two iterated learning procedures will show a loss of accuracy at the beginning of each generation. That is because one or both agents are reset, and are not able to completely re-learn from the data kept from the previous generation during the pre-training phase. However, at the end of each generation, all these algorithms can ensure a perfect game performance.

While the use of NIL has little effect on the game performance, given a sufficient number of rounds, these procedures have a clear positive effect on topological similarity. In Figure 3-(b), we can see that the no-reset case has the lowest average $\rho$ while the iterated learning cases all have higher means (and increasing). We provide extra experiments in Appendix E, which demonstrate the robustness of NIL under different scenarios. The discussion on the specific impact of each agent and why the reset-Alice and reset-Bob behave differently is in Section 5.

### 4.3 HIGH TOPOLOGICAL SIMILARITY AND INTERVAL OF ADVANTAGE

In this section, we explore further the phenomenon caused by the learning speed advantage on NIL. From the discussion in section 3.1 and the experimental results in section 4.1, we know that $I_a$ and $I_b$ play an important role in NIL: they should not be too large nor too small. Intuitively, if $I_a$ is too small, Alice will learn nothing from the previous generation, hence the NIL amounts to playing

| $I_a$ | 100 | 200 | 400 | 800 | **1200** | **1500** | **2000** | 3000 | 5000 | 8000 |
|---|---|---|---|---|---|---|---|---|---|---|
| $\mathbb{E}[r_{71:80}]$ | 0.293 | 0.828 | 0.928 | 0.951 | 0.958 | 0.961 | 0.952 | 0.956 | 0.955 | 0.949 |
| $\mathbb{E}[\rho_{1:10}]$ | 0.225 | 0.429 | 0.452 | 0.483 | **0.556** | **0.575** | **0.566** | 0.494 | 0.481 | 0.443 |
| $\mathbb{E}[\rho_{71:80}]$ | 0.203 | 0.706 | 0.836 | 0.886 | 0.899 | 0.935 | 0.936 | 0.929 | 0.889 | 0.837 |
| $I_b$ | 10 | 20 | 40 | 80 | **120** | **160** | **200** | 300 | 400 | 800 |
| $\mathbb{E}[r_{71:80}]$ | 0.954 | 0.946 | 0.961 | 0.954 | 0.962 | 0.959 | 0.962 | 0.957 | 0.961 | 0.944 |
| $\mathbb{E}[\rho_{1:10}]$ | 0.415 | 0.381 | 0.488 | 0.496 | **0.591** | **0.535** | **0.557** | 0.498 | 0.488 | 0.448 |
| $\mathbb{E}[\rho_{71:80}]$ | 0.927 | 0.937 | 0.929 | 0.928 | 0.936 | 0.891 | 0.888 | 0.897 | 0.891 | 0.880 |

Table 1: Values of 3 metrics when varying $I_a$ or $I_b$, highlighting an interval where the topological similarity grows high.

only **one** interacting phase. If $I_a$ is too large, from the trend in Figure 2-(c), we may expect that the increase of expected $\rho$ should be small in each generation, because Alice will perfectly learn $D_i$, and hence have a $\rho$ similar to its predecessor. Hence we speculate that the value of $I_a$ should have a "bottleneck" effect, i.e., a too large one or a too small one will both harm the performance of NIL. A similar argument can also applied in the selection of $I_b$.

To verify our argument, we run NIL with different values of $I_a$ and $I_b$, examining the behavior of the following 3 different quantitive metrics:

- $\mathbb{E}[r_{71:80}]$: The average reward of the last ten generations (game performance);
- $\mathbb{E}[\rho_{1:10}]$: The average value of $\rho$ for the first ten generations (converging speed);
- $\mathbb{E}[\rho_{71:80}]$: The average value of $\rho$ for the last ten generations (converged $\rho$).

From the results presented in Table 1, we can see the importance of the number of pre-training rounds not being too large nor too small. The suitable $I_a$ and $I_b$ are shown in bold. Furthermore, combining Figure 2 and Table 1, the interval of suitable $I_a$ lies between 1000 to 2000 while it lies between 100 to 200 for $I_b$, which provides us an effective way to choosing hyper-parameters.

### 4.4 TOPOLOGICAL SIMILARITY AND VALIDATION PERFORMANCE

In this last series of experiments, we aim to explore the relationship between topological similarity and the generalisation ability of our neural agents, which can also indirectly reflect the expressivity of a language. We measure this ability by looking at their validation game performance: we restrict the training examples to a limited numbers of objects (i.e., the training set), and look at how good are the agents at playing the game on the others (i.e., the validation set). Figure 4-(a) demonstrates the strength of the iterated learning procedure in a validation setting. To illustrate the relationship between $\rho$ and validation performance, we randomly choose $I_a \in [60, 4000]$ and $I_b \in [5, 200]$ and conduct a series of experiments. Those for which $I_a$ and $I_b$ are not in their optimal range will yield a worse performance on both validation test and topological similarity. In Figure 4-(b), we record the results from different experimental settings and draw the zero-shot performance given the topological similarity of the emergent language. This shows the linear correlation between these two metrics, and a significance test confirms it: the correlation coefficient is 0.928, and the associated $p$-value is $3.8 * 10^{-104}$. Hence, under various experimental settings, the validation performance and the topological similarity are strongly correlated. Table 2 shows that when the size of validation set increases, using iterated learning can always improve the validation performance: in all the cases, both-reset algorithm always yields the best performance. The fact that the Alice-reset setting performs better than the Bob-reset setting also matches our analysis well.

## 5 DISCUSSION: A PARALLEL WITH LANGUAGE EVOLUTION

We can observe an interesting phenomenon in Figure 3-(b):[2] the topological similarity of the emergent language always increases at first, whether we use iterated learning or not. This is akin to the effect apparent for $\rho$ in Figure 2-(c): continuing training will imply fine-tuning to examples that

---

[2]This can be viewed in more details when looking at the probabilistic analysis presented in Appendix D.

| Valid set size | 0 | | 8 | | 16 | | 32 | |
|---|---|---|---|---|---|---|---|---|
| | Train | Valid | Train | Valid | Train | Valid | Train | Valid |
| No-reset | 0.985 | - | 0.986 | 0.136 | 0.990 | 0.132 | 0.995 | 0.102 |
| Bob-reset | 0.967 | - | 0.943 | 0.094 | 0.962 | 0.152 | 0.947 | 0.116 |
| Alice-reset | 0.981 | - | 0.976 | 0.598 | 0.979 | 0.280 | 0.947 | 0.210 |
| Both-reset | 0.988 | - | 0.986 | 0.847 | 0.984 | 0.755 | 0.973 | 0.558 |

Table 2: Validation performance under different validation set sizes.

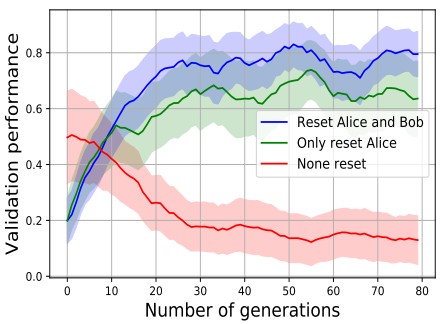

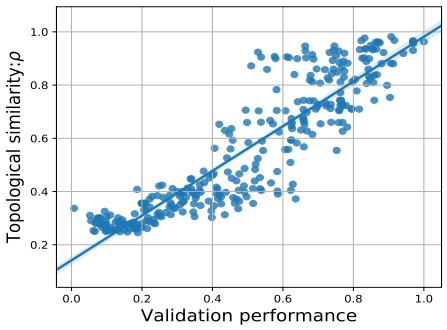

(a) Validation performance, validation set size is 8.    (b) Results of validation performance and $\rho$.

Figure 4: Validation performance and topological similarity with validation size equals eight. NIL leads to the evolution of languages which allow agents to perform well on unseen items.

are not necessarily of good quality. However, through the generational resets and limited number of pre-training examples, iterated learning allows small generational improvements: this is because constraining the agent to learn with smaller amounts of data at each generation — through a 'bottleneck' (Kirby & Hurford, 2002) — forces the emergence of a more structured language. This limitation on the amounts of data available corresponds in our algorithm to limiting the number of pre-training rounds of the agents, to a number in what we denoted as the 'interval of advantage'. In NIL, we use the weak pre-training to simulate this bottleneck, and achieve a good result: the values of $I_a$ and $I_b$ have an effect similar to the bottleneck studied in (Kirby et al., 2015) (more details are provided in Appendix D). Extending this parallel with the evolution of natural language, we can relate the learning speed advantage provided by high-$\rho$ languages to the speaking agent to the *compressibility pressure* (Kirby et al., 2015), and the better ability to generalize provided by high-$\rho$ languages to the listening agent to the *expressivity pressure* (Kirby et al., 2015).

This comparison allows us to address one important difference between our neural iterated learning algorithm and the original version: our speaking and listening agents are not identical. Actually, the speaking module and listening module of human are also not identical, but the works on traditional iterated learning do not pay much attention to such differences. From Figure 3-(b) and Figure 4-(a), it is clear that Alice and Bob are affected differently by the generational resets, and thus do not offer the same contribution to the final performance.[3] From this parallel, we retain that iterated learning is also linked to the emergence of a certain form of compositionality when applied to neural agents. Besides, we believe that the correlation between topological similarity and validation performance that we highlight in Section 4.4 is another argument in favor of a relationship between compositionality and generalization, which has recently been explored (Kottur et al., 2017; Choi et al., 2018; Andreas, 2019).

---

[3]However, this parallel may not explain how differently they contribute to gains in topological similarity, since we must factor in the differences between their pre-training procedures, and especially the fact that Alice is pre-trained by minimizing cross-entropy.

## 6 CONCLUSION

In this paper, we find and articulate the existence of the learning speed advantages offered by high topological similarity, with which, we propose the NIL algorithm to encourage the dominance of high compositional language in a multi-agent communication game. We show that our procedure, consisting in resetting neural agents playing a referential game and pre-training them on data generated by their predecessors, can incrementally advantage emergent languages with high topological similarity. We demonstrate its interest by obtaining large performance improvements in a validation setting, linking compositionality and ability to generalize to new examples. The robustness of the algorithm is also verified in various experimental settings. Finally, we hope the proposed probabilistic model of NIL could inspire the application of NIL in more complex neural-agents-based systems.

## ACKNOWLEDGEMENT

We show our sincere gratitude to Kenny Smith, Ivan Titov, Stella Frank and Serhii Havrylov for their helpful discussion and comments that greatly improved the manuscript.

We would also like to thank the members from Prof. Jun Zhao's team at Institute of Automation, Chinese Academy of Sciences, e.g. Dr. Kang Liu, Xiang Zhang and Xinyu Zuo, for sharing computing resources to run some experiments as well as sharing their pearls of wisdom with us during the course of this research, and we thank 3 anonymous reviewers for their insights and comments.

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

APPENDIX A: PARAMETER SETTINGS

Unless specifically stated, the experiments mentioned in this paper use the hyper-parameters given in Table 3. The code is available at `https://github.com/Joshua-Ren/Neural_Iterated_Learning`.

| Notation | Value | Description |
|---|---|---|
| $N_a$ | 2 | Number of all attributes |
| $N_v$ | 8 | Number of possible values for each attribute |
| $N_L$ | 2, 3 | Message length |
| $|V|$ | 8+[0,4,8,16,32,64] | Vocabulary size. |
| $I$ | 80, 100 | Maximum number of generations |
| $I_a$ | $\geq 100, \leq 8000$ | Maximum pre-train **rounds** for Alice |
| $I_b$ | $\geq 10, \leq 800$ | Maximum pre-train **batches** for Bob |
| $I_g$ | $\geq 100, \leq 8000$ | Maximum interacting rounds |
| $I_s$ | 10, 100, 1000 | Maximum rounds for transmitting phase |
| $N_h$ | 128 | Hidden layer size |
| $N_b$ | 64 | Batch size |
| $c$ | 2, 5, 15, 30 | Number of candidates (including the target) |
| $lr$ | $\geq 10^{-5}, \leq 10^{-3}$ | Learning rate |

Table 3: Value of hyper-parameters.

APPENDIX B: DIFFERENT TYPES OF LANGUAGES: A TOY EXAMPLE

| Group | Compsitional (8) | Holistic (16) | Other (232) |
|---|---|---|---|
| | *blue box = aa* | *blue box = ba* | *blue box = aa* |
| Language | *red box = ba* | *red box = aa* | *red box = bb* |
| Examples | *blue circle = ab* | *blue circle = ab* | *blue circle = aa* |
| | *red circle = bb* | *red circle = bb* | *red circle = bb* |
| $\rho$ | 1 | 0.5 | 0.1$\sim$0.7 |

Table 4: Different groups of language and their topological similarity.

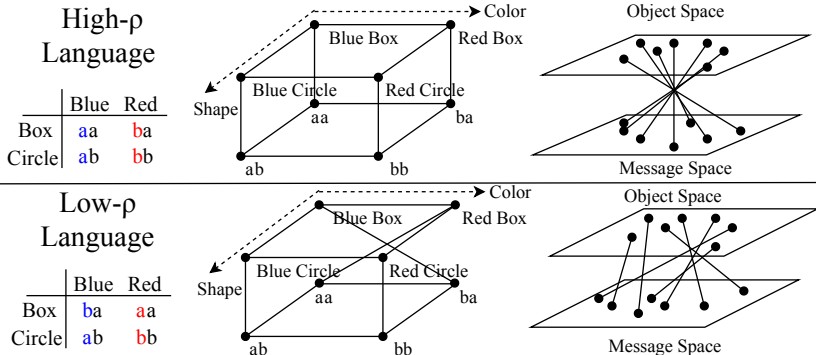

Figure 5: A simple representation of two languages corresponding to topological similarities of $\rho = 1$ (top) and $\rho = 0.5$ (bottom).

To better understand how topological similarity can measure the compositionality of one language, and to give some intuitions on how languages having different $\rho$ would like, we provide and illustrate a toy example in this appendix. In this example, the object space is $\mathcal{X} = \{$blue box, blue circle, red box, red circle$\}$ and the message space is $\mathcal{M} = \{aa, ab, ba, bb\}$. Any

set of mappings from four distinct objects to four messages (not necessarily distinct, i.e. same message could correspond to different objects) forms a language. Hence, there exist $4^4 = 256$ possible languages in this toy example. Following the principles provided in (Kirby et al., 2015), we define the following concepts for describing a language:

- **Unambiguous language**. A type of language that can unambiguously describe all objects in $\mathcal{X}$. In other words, the mappings between $\mathcal{X}$ and $\mathcal{M}$ are bijectional. In this example, there exist $4 \times 3 \times 2 \times 1 = 24$ such languages.

- **Compositional language**. A type of unambiguous language that exhibits systematic compositional structure when forming messages. Such languages can use different symbols to represent different attributes of meaning and combine these symbols in a systematic way to form a message such that the meaning of the whole message is formed from a simple combination of the meaning of its parts. For example, following the rules of $S \rightarrow XY$, and $X : blue \rightarrow b; X : red :\rightarrow b; Y : box \rightarrow a; X : circle :\rightarrow b$, we can derive a compositional language like the example in Table 4. In this example, we have $4 + 4 = 8$ such languages.

- **Holistic language**. A type of unambiguous language but does not exhibits full systematic structures. In other words, holistic languages are those unambiguous language who are not compositional languages. In this example, we have $24 - 8 = 16$ such languages.

- **Degenerate language**. A type of ambiguous language that maps all objects to the same message. In this example, we have $4$ such languages.

- **Degenerate component**. Any ambiguous language having degenerate component, i.e., there may be more than one objects mapping to the same message. The existence of degenerate component makes the language ambiguous.

Note that the number of unambiguous languages is usually much smaller than that of ambiguous languages, and the number of compositional languages is usually smaller than that of holistic languages. Using permutation and combination, we can calculate the numbers of all possible languages, unambiguous language, compositional language and holistic language as:

$$\# \text{ all possible languages} = \left(|V|^{N_L}\right)^{\left(N_v^{N_a}\right)} \tag{4}$$

$$\# \text{ unambiguous languages} = \frac{(|V|^{N_L})!}{(|V|^{N_L} - N_v^{N_a})!} \tag{5}$$

$$\# \text{ compositional languages} = \frac{N_L!}{(N_L - N_a)!} \cdot \left(\frac{|V|!}{(|V| - N_v)!}\right)^{N_a} \tag{6}$$

$$\# \text{ holistic languages} = \# \text{ unambiguous languages} - \# \text{ compositional languages} \tag{7}$$

From the above equation, it is easy to see that the gap between the number of compositional languages and holistic languages would become larger when $N_v$, $N_a$, $N_L$ and $|V|$ increase. Further, this means that it becomes even harder to pick a compositional language when randomly sample a language. That could explain why the expected topological similarity of the emergent language may increase when smaller $N_L$ and $|V|$ are applied, as illustrated in (Lazaridou et al., 2018; Cogswell et al., 2019).

Besides the numbers of different languages, another key difference among these languages is the topological similarity (i.e., $\rho$), as illustrated in section 2.3. As the language studied in this paper is defined as a mapping function from a meaning (i.e., an object) to a message, a compositional language must ensure that the meaning of a symbol is a function of the meaning of its parts. In other words, compositional languages are neighborhood related: nearby meanings tend to be mapped to nearby signals. Or to say, nearby meanings that share similar attributes are likely to share similar message symbols (Brighton & Kirby, 2006). Thus, as the difference between messages are measured by edit distance, the compositional languages will have a higher $\rho$ than the holistic ones. However, the existence of degenerate component also change the value of $\rho$: the $\rho$ of a degenerate language might be higher than that of a holistic language.

From the above discussions, we find that making the highly compositional languages dominate is a challenging task: it occupies a really small portion among all possible languages, and only using

topological similarity also cannot tell them apart from those who are highly degenerate. However, the proposed algorithm can solve this problem almost perfectly: it uses the learning speed advantage caused by high topological similarity to increase the posterior probability of high-$\rho$ languages, and uses the interacting phase to rule out the degenerate components. The details of how the probability of languages changes in different phase of our algorithm are illustrated in Appendix C.

## APPENDIX C: PROBABILISTIC MODEL OF THE SYSTEM

**Probabilistic Model of Emergent Languages:**

In section 2.3, we define a language as a mapping function from object space $\mathcal{X}$ to the message space $\mathcal{M}$, i.e., $\mathcal{L}(\cdot) : \mathcal{X} \mapsto \mathcal{M}$. Here we discuss how to describe the probability of a specific language, i.e., $P(\mathcal{L})$.

Suppose that we have $N$ possible different objects $(x_1, x_2, ..., x_N)$, where $N = N_v^{N_a}$, and the messages are conditionally independent given an object $x_n$ (where $n \in [1, 2, \ldots, N]$), i.e.:

$$P(\mathcal{L}) = P(\mathbf{m_1}, ..., \mathbf{m_N}|x_1, ..., x_N) = \prod_{n=1}^{N} P(\mathbf{m_n}|x_1, ..., x_N) = \prod_{n=1}^{N} P(\mathbf{m_n}|x_n). \qquad (8)$$

Assume that messages are uniformly sampled from $\mathcal{M}$ whose size is $M = |V|^{N_L}$, we could have $P(\mathbf{m_n}|x_n) = \frac{1}{M}, \forall n \in \{1, 2, ..., N\}$. Hence the initial probability (or prior probability) of any possible language is $\left(\frac{1}{M}\right)^N$. We define the posterior distribution of languages as the distribution after our neural iterated learning algorithm (NIL), i.e. $P(\mathcal{L}|\text{NIL})$.

Then, our goal is to enhance the posterior probability of the high-$\rho$ languages, which is equivelant to enhance the expectation of $\rho$, i.e.:

$$\mathbb{E}_{\mathcal{L} \sim P(\mathcal{L}|\text{NIL})}[\rho(\mathcal{L})] = \sum_i \rho(\mathcal{L}_i)P(\mathcal{L}_i|\text{NIL}). \qquad (9)$$

It is obvious that $\mathbb{E}_{\mathcal{L}}[\rho(\mathcal{L})]$, the expected topological similarity of languages following the prior probability, is quite low, as the high-$\rho$ languages only occupy an extremely small fraction.

**Definition of the Agents:**

Following the structure provided in Figure 1, we define the speaking agent (Alice) and listening agent (Bob) formally here.

Alice is a bunch of neural networks that can map any input object $x$ to a discrete message $\mathbf{m}$. So we define it as $\mathbf{m} = h(x), h : \mathcal{X} \mapsto \mathcal{M}$. As Alice generate discrete messages with softmax layers, the probabilistic distribution of different words in $\mathbf{m}_n$ can be obtained. In the example provided in Figure 1, we can have $P(m_1|x)$ and $P(m_2|x, m_1)$ by reading the distribution from softmax layers. In more general cases, we could obtain $P(m_l|x, m_{l-1}, m_{l-2}, \dots)$ following the same method. Thus, we can directly calculate the probability of specific $\mathbf{m}$ given $x$ for Alice as follow:

$$P(\mathbf{m}|x) = P(m_1|x) \prod_{l=2}^{N_L} P(m_l|x, m_{l-1}, m_{l-2}, \dots). \qquad (10)$$

If we feed all possible $x$ to Alice and calculate the corresponding $P(\mathbf{m}|x)$, we then could calculate the probability distribution of all languages after training Alice, following equation (8) and (9). Then, we can state our goal as to obtain a high $\mathbb{E}_{\mathcal{L} \sim P(\mathcal{L}|\text{NIL})}[\rho(\mathcal{L})]$ by using NIL to update the parameters of the neural network.

In our setting, the posterior probability of languages is decided by Alice with its softmax layers. Bob plays a role of assistant to ensure the robustness of NIL, which will be further illustrated in Appendix D and E. From Figure 1, we could see that the inputs of Bob are a discrete message $\mathbf{m}$ and $c$ different objects. As Bob will calculate a score $s_c$ for each object $c_c$, we can denote its function as $s = f(\mathbf{m}, x), f : \mathcal{M} \times \mathcal{X} \mapsto \mathbb{R}^1$.

**Probabilistic Description of Language Evolution in NIL:**

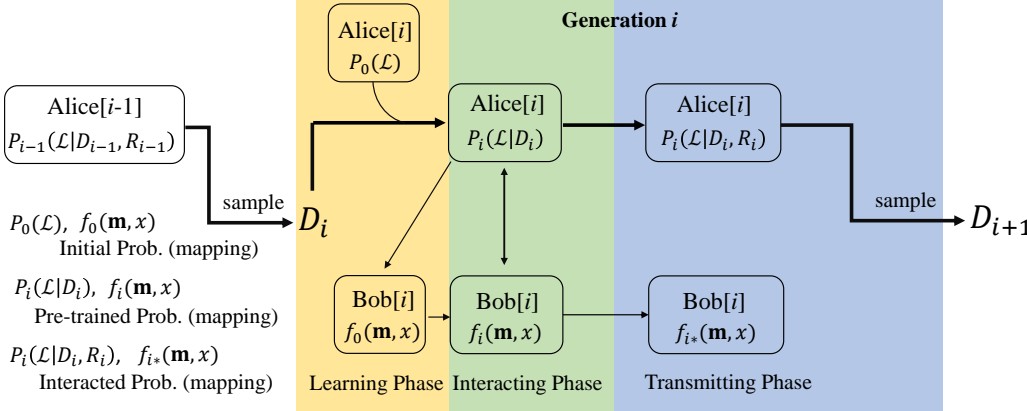

Figure 6: Probabilitic explanation of different phases in NIL.

To avoid confusion, we specify all the probabilities involved in NIL in the left corner of Figure 6. In the figure, the shadow regions with different colors represent the three phases of NIL in **ONE** generation. Thus, one generation of NIL could be described as:

1. **Initialization**: At the beginning of generation $t$, the initial probability of Alice[i] is $P_0(\mathcal{L})$, which is same as the prior probability of $P(\mathcal{L})$ mentioned before, as Alice[i] is always randomly initialized. The initial function of Bob[i] is represented as $f_0(\mathbf{m}, x)$.

2. **Learning Phase**: Then following Algorithm 1, Alice[i] will be pre-trained using the data sampled from the previous generation, i.e. $D_i$. The pre-trained probability of languages is defined as $P_i(\mathcal{L}|D_i)$. Bob[i] will then be pre-trained using the sample generated by $P_i(\mathcal{L}|D_i)$, using REINFORCE procedure, after which, its function becomes $f_i(\mathbf{m}, x)$.

3. **Interacting Phase**: The pre-trained Alice[i] and Bob[i] then interact and update their parameters together following the REINFORCE procedure described in section 3.2. In each round of the game, Alice[i] would first use argmax to select $\mathbf{m}$ with the highest probability given a randomly selected object $x$, both agents would then update their parameters if $R = 1$, i.e. the data pair $\langle \mathbf{m}, x \rangle$ could assist them to accomplish the referential game successfully. We argure that this process has the same effect as the following procedure: we first sample a data set $D_* \sim P_i(\mathcal{L}|D_i)$, and then delete the data pairs that cannot unambiguously deliver information to form a refined data set $R_i$. Then, the interacted probability of Alice[i] can be represented by $P_i(\mathcal{L}|D_i, R_i)$. As Bob also update its parameters in this phase, we define its interacted function as $f_{i*}(\mathbf{m}, x)$.

4. **Transmitting Phase**: Finally, in the transmitting phase, we sample $D_{i+1} \sim P_i(\mathcal{L}|D_i, R_i)$ by: i)randomly feeding $x_n$ to Alice[i]; ii) sample a message $\mathbf{m}_n \sim P_i(\mathbf{m}|x_n, D_i, R_i)$. Note that Bob[i] is not involved in this phase.

From all sections above, we argue that Alice plays an important role in all the phases in NIL while Bob only helps to make the languages effective during interaction phases. As we will discuss the role of Alice and Bob in further details in Appendix E, we only provide an intuition of how the language changes in NIL in the following paragraphs.

Overall, the objective of our NIL design is to ensure the expected topological similarity of emergent languages would increase over generations, as expressed by equation (3). As the languages with higher $\rho$ would be learned faster, which is stated as Hypothesis 1, we can expect those high-$\rho$ languages to have a higher pre-trained probability in $P(\mathcal{L}|D_i)$ than in $D_i$, i.e.:[4]

$$\mathbb{E}_{\mathcal{L} \sim P(\mathcal{L}|D_i)}[\rho(\mathcal{L})] \geq \mathbb{E}_{\mathcal{L} \sim D_i}[\rho(\mathcal{L})]. \tag{11}$$

Note that this inequality is not a strict corollary, but it is very likely to hold as long as we have an appropriate $I_a$. In the worst case, we can chose an extremely large $I_a$ to make Alice learn $D_i$ perfectly. However, we could verify it by the experimental results as well as the explanation in Appendix D that the weak pre-training can indeed help us to achieve a higher expected $\rho$. Then, in the interacting phase, we may expect:

$$\mathbb{E}_{\mathcal{L} \sim P(\mathcal{L}|D_i, R_i)}[\rho(\mathcal{L})] = \mathbb{E}_{\mathcal{L} \sim P(\mathcal{L}|D_i)}[\rho(\mathcal{L})], \tag{12}$$

as the compositional languages and holistic languages are both unambiguous and the game performance cannot tell them apart. Finally, during the transmitting phase, we have $D_{i+1} \sim P_i(\mathcal{L}|D_i, R_i)$. Assuming that we sampled enough $D_{i+1}$ to ensure it has a very similar distribution to $P_i(\mathcal{L}|D_i, R_i)$, it is reasonable to have:

$$\mathbb{E}_{\mathcal{L} \sim D_{i+1}}[\rho(\mathcal{L})] = \mathbb{E}_{\mathcal{L} \sim P(\mathcal{L}|D_i, R_i)}[\rho(\mathcal{L})]. \tag{13}$$

Sum up from the above, equation (3) can be obtained by combining equation (11-13).

## APPENDIX D: MORE ON THE LEARNING SPEED ADVANTAGE

*Amplifying mechanism* and *learning speed advantage* are the two main elements for the success of NIL. The former on is elaborated in section 3.2 and Appendix C, under the assumption that the learning speed advantage of high-$\rho$ language indeed exist. In this section, we will explain why such an advantage exist by experimental results and a toy example.

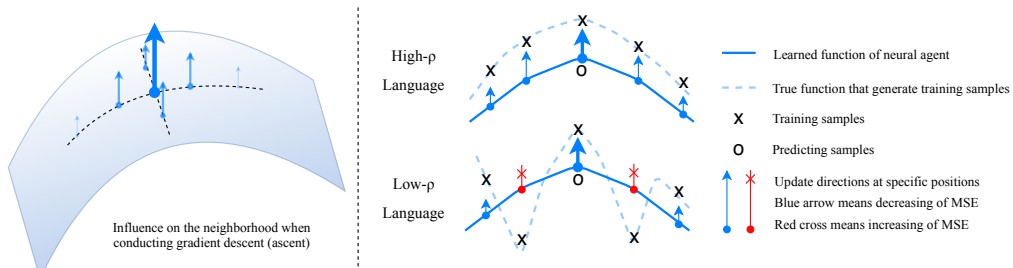

Figure 7: Illustration of learning a high-$\rho$ language and low-$\rho$ language.

**Example for Supporting Hypothesis 1:**

This hypothesis claims that a high-$\rho$ language would be leared faster than a low-$\rho$ one on the speaker side. As we can directly represent the posterior probability of any language from Alice's perspective, the assertion of *"learned faster"* can be converted to *"the posterior probability increases faster"*. We use a toy example, i.e. two languages in Table 4, to demonstrate how such an advantage emerges and how it works. To make the notation concise, we use "BB, RB, BC, RC" to represent "blue box, red box, blue circle, red circle" respectively. The probability of the compositional language and the holistic language in Table 4 can be represented as:

---

[4]Figure 2-(c) might be a good example

$$P(\mathcal{L}_{\text{cmp}}) = P(m_1{=}a|BB) \cdot P(m_2{=}a|BB, m_1{=}a) \cdot P(m_1{=}b|RB) \cdot P(m_2{=}a|RB, m_1{=}b) \cdot C \quad (14)$$

$$P(\mathcal{L}_{\text{hol}}) = \underbrace{P(m_1{=}b|BB)}_{①} \cdot \underbrace{P(m_2{=}a|BB, m_1{=}b)}_{②} \cdot \underbrace{P(m_1{=}a|RB)}_{③} \cdot \underbrace{P(m_2{=}a|RB, m_1{=}a)}_{④} \cdot C \quad (15)$$

$$C = \underbrace{P(m_1{=}a|BC)}_{⑤} \cdot \underbrace{P(m_2{=}b|BC, m_1{=}a)}_{⑥} \cdot \underbrace{P(m_1{=}b|RC)}_{⑦} \cdot \underbrace{P(m_2{=}b|RC, m_1{=}b)}_{⑧} \quad (16)$$

where $C$ is the common part for both languages.

As we are using stochastic gradient descent algorithm to update the parameters of Alice, it straightforward to see that the update of gradient from one point will 'pull up' the neighbourhood region of function $h$, which is shown in the left panel of Figure 7. Then, we can speculate that if one data sample belonging to both the two language comes, e.g. $\langle ab, BC \rangle$, the following probabilities would increase at the same time:

$$P(m_1{=}a|BC); \quad P(m_1{=}a|BB); \quad P(m_1{=}a|RC), \quad (17)$$

as the input of them are similar with $BC$ (only one attribute changes). As the conditional probabilities must sum to 1, the following probabilities would decrease:

$$P(m_1{=}b|BC); \quad P(m_1{=}b|BB); \quad P(m_1{=}b|RC). \quad (18)$$

Thus, when Alice learns the data sample $\langle ab, BC \rangle$, $P(\mathcal{L}_{\text{cmp}})$ may have two terms increased, i.e., terms ⑤ and ① . For $P(\mathcal{L}_{\text{hol}})$, however, the decrease of term ① will harm the increase of term ⑤ , hence $P(\mathcal{L}_{\text{hol}})$ increases slower than $P(\mathcal{L}_{\text{cmp}})$ (The fact that term ⑦ decreases on both sides would not change our deduction).

**Example for Supporting Hypothesis 2:**

We can use a similar explanation for the advantage at Bob. Recall that Bob is defined as a mapping function $f$ from $\mathcal{M} \times \mathcal{X}$ to $\mathbb{R}^1$. Following the principle mentioned above, if Bob learns $\langle ab, BB \rangle$, a bunch of function values would increase, i.e. $f(ab, BB)$, $f(aa, BB)$, $f(bb, BB)$, $f(ab, BC)$, and $f(ab, CB)$, as they are all close to each other in the input space. Then it is easy to find that two terms in the compositional language in Table 4 are increased while only one term increases in the holistic language. That is, the score of high-$\rho$ language would increases faster.

We can also think hypothesis 2 in the following way. With the intuition that a language with higher $\rho$ tends to be smoother and to have fewer inflection points than one with lower $\rho$, the learning speed advantage given by highly compositional languages can be illustrated by the example provided in Figure 7. In the example, language is considered to be a one-dimensional mapping function, which is represented by the dotted lines in Figure 7. The object-message pairs, which are represented by the cross marks, are the points that satisfy the mapping function. The solid line represents the mapping function of the learning agent. Suppose the target output (i.e. the third cross mark in each figure) is larger than the predicting output (i.e. the circle mark), the optimizer will update the parameters of the neural network following the direction of the gradient, as illustrated by the bold arrows in the figure. Such an update will also pull the neighbouring parts of the function up, as illustrated by the smaller arrows on the solid curve.

The smoothness of high-$\rho$ languages implies that the MSE of neighbouring positions will also be reduced by this update, while the MSE of neighbors would be increased in the case of a low-$\rho$ language. Such a trend is represented by the blue arrows and red crossed-arrows in Figure 7: the blue one means a decrease of the MSE at the specific position while the red one means increases of MSE. In other words, for a high-$\rho$ languages, an update corresponding to one data sample is likely to have a larger positive effect on other data samples, and hence ensure a higher learning speed. Meanwhile, for a low-$\rho$ language, one data sample would have both positive and negative effects on its neighbors and thus lead to a lower learning speed.

## APPENDIX E: ROBUSTNESS OF NIL

In this section, we will provide experimental results to demonstrate the robustness of the proposed method. The influence of hyperparameters (e.g. vocabulary size, message length) as well as the role played by Alice and Bob are both elaborated.

**Robustness for Hyperparameters on Message Space:**

The message space are decided by the vocabulary size $|V|$ and the message length $N_L$. Thus, we first make experiments to see the effects of different $|V|$ and $N_L$ on $\mathbb{E}_{\mathcal{L}\sim P(\mathcal{L}|\text{NIL})}[\rho(\mathcal{L})]$.

From the discussion in Appendix B, we know that when $|V|$ and $N_L$ are large, making high-$\rho$ language dominate in the posterior probability is very hard, as the compositional languages only occupy an extremely small portion. Such a trend could also be found in Table 5, as the finally converged expectation of topological similarity becomes lower with larger $|V|$ or $N_L$.

Our algorithm, however, is very robust to different values of $|V|$ and $N_L$. By comparing different columns in Table 5, $\mathbb{E}_{\mathcal{L}\sim P(\mathcal{L}|\text{NIL})}[\rho(\mathcal{L})]$ decreases very slow with the increasement of $|V|$ and $N_L$. An extreme example is that, the converged $\rho$ can still be roughly 0.8 with $|V| = 72$. The performance of validation accuracy seems more robust when $|V|$ and $N_L$ changes: the NIL can always obtain more than 80% accuracy compared to the none reset case (roughly 15%).

Furthermore, compared with $|V|$, $N_L$ has a stronger impact on the performance in terms of all metrics but the validation performance, as it is shown in Table 5 that the performance with $N_L = 3$ is significantly lower than its counterpart when $N_L = 2$. One possible explanation is that the increasing of $N_L$ brings an exponential change to the message space. However, no matter how $|V|$ and $N_L$ change, $\mathbb{E}_{\mathcal{L}\sim P(\mathcal{L}|\text{NIL})}[\rho(\mathcal{L})]$ is always significantly higher that the compositionality of emergent languages given by baseline model, i.e. 0.3.

| | $N_L$ | $|V|=8$ | $|V|=12$ | $|V|=16$ | $|V|=24$ | $|V|=40$ | $|V|=72$ |
|---|---|---|---|---|---|---|---|
| $\mathbb{E}[\rho_{71:80}]$ | 2 | 0.986±0.01 | 0.937±0.02 | 0.933±0.01 | 0.854±0.02 | 0.830±0.02 | 0.793±0.02 |
| | 3 | 0.712±0.01 | 0.833±0.01 | 0.798±0.02 | 0.777±0.01 | 0.793±0.02 | 0.780±0.03 |
| $\mathbb{E}[\rho_{1:10}]$ | 2 | 0.767±0.18 | 0.690±0.18 | 0.684±0.20 | 0.630±0.17 | 0.668±0.19 | 0.572±0.14 |
| | 3 | 0.528±0.11 | 0.647±0.15 | 0.640±0.17 | 0.664±0.14 | 0.637±0.16 | 0.628±0.21 |
| $G_{0.85}$ | 2 | 9 | 16 | 10 | 37 | 68 | - |
| | 3 | - | - | 39 | - | - | 59 |
| Valid Acc. | 2 | 0.868±0.14 | 0.914±0.06 | 0.833±0.11 | 0.866±0.11 | 0.801±0.10 | 0.828±0.14 |
| | 3 | 0.804±0.13 | 0.677±0.16 | 0.773±0.15 | 0.858±0.10 | 0.867±0.01 | 0.900±0.07 |

Table 5: Values of 4 metrics when $|V|$ and $N_L$ changes. Metric $G_{0.85}$ means the first generation that the average $\rho$ of the previous three generations exceed 0.85. The notation "-" means the agents never satisfy the requirement.

**Robustness on Degenerate Components:**

From the discussions in Appendix B, we know that the $\rho$ of a language who has many degenerate components will also be high, and hence can be learned faster by Alice in the learning phase. Thus, it is necessary to check whether our algorithm can avoid the mode collapse caused by the degenerate components. Intuitively, the degenerate components can be filtered out during the interacting phase, as the REINFORCE algorithm ensure that the parameters of the agent will only be updated with respect to $R = 1$, i.e. the language is effective and thus unambiguous.

To verify our hypothesis, we first observe how the number of message types, i.e. the number of different messages used to describe all 64 objects, changes during NIL. It is straightforward to see that a language without any degenerate component would have 64 different message types. As shown in Figure 8, all methods could achieve high numbers if message types, which indicates that the REINFORCE algorithm could always filter out the degenerate components efficiently.

Furthermore, we design two challenging tasks for NIL:

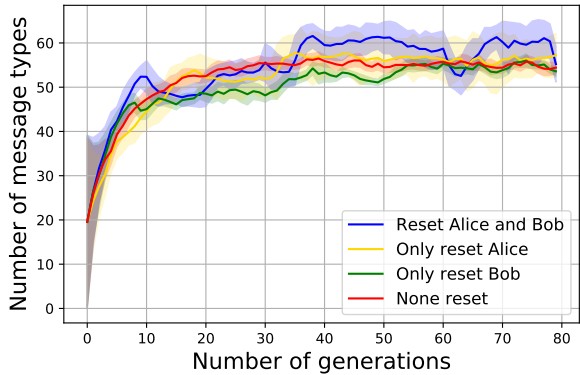

Figure 8: Numbers of message types from different settings.

1. **Degenerate initialized:** We let Alice learn from a pure degenerate language at the beginning of each generation, before it learns from $D_i$.

2. **Degenerate mixed:** We mix the data pair generated by a pure degenerate language to $D_i$ and ensures the proportion of the degenerate pairs is more than $50\%$, which makes Alice easier to collapse to a degenerate language during learning phase.

We then compare the performance, i.e. the expected $\rho$ and validation accuracy, of agents in different tasks. The results shown in Figure 9 demonstrate that our NIL is very robust to the influence of degenerate component, as both $\mathbb{E}_{\mathcal{L}\sim P(\mathcal{L}|\mathrm{NIL})}[\rho(\mathcal{L})]$ and the validation score are much higher than the none reset baseline's performance.

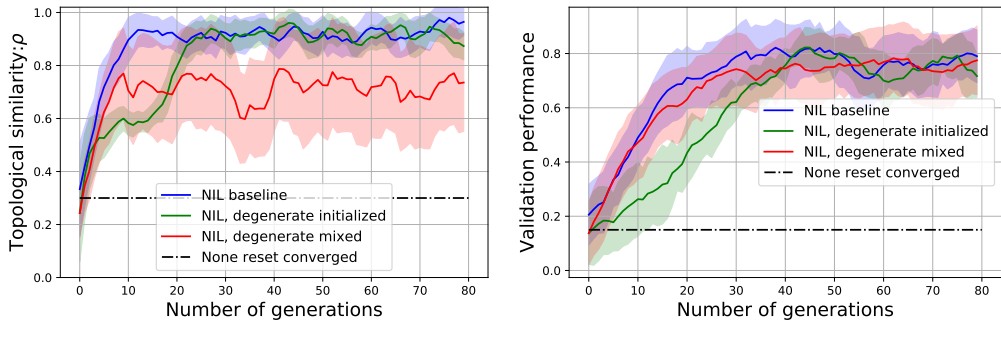

(a) Performance on topological similarity

(b) Performance on validation accuracy.

Figure 9: Two corner case test. NIL with degenerate initialized means Alice is initialized with a degenerate language at the beginning of each generation. NIL with degenerate mixed means Alice is initialized with a degenerate language, AND the $D_i$ is mixed with $I_s$ degenerate language pairs.

**The Role of Bob's Pre-training**

From the discussions above, it is easy to understand why $\mathbb{E}_{\mathcal{L}\sim P(\mathcal{L}|\mathrm{NIL})}[\rho(\mathcal{L})]$ would gradually increase in NIL and how the REINFORCE applied in interacting phase can filter the degenerate component. However, the role played by Bob, especially in the learning phase where Bob only update its own parameters, is not straightforward. In short, the pre-training of Bob makes the algorithm more robust, especially at the beginning of the interacting phase. We record the value of $\mathbb{E}_{\mathcal{L}\sim P(\mathcal{L}|\mathrm{NIL})}[\rho(\mathcal{L})]$ every 20 iterations among learning phase and interacting phase, and plot the results of two generations in Figure 10.

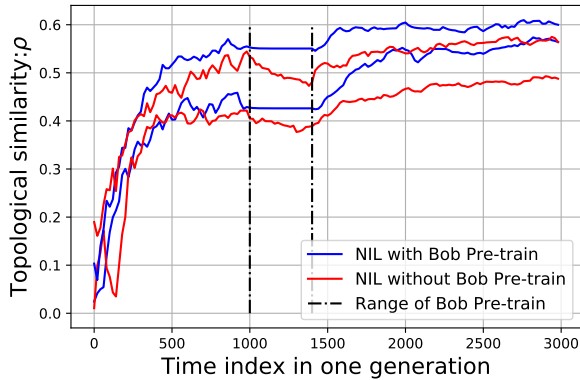

Figure 10: The change of $\mathbb{E}_{\mathcal{L}\sim P(\mathcal{L}|\mathrm{NIL})}[\rho(\mathcal{L})]$ in generation 3 and 6.

In this figure, the x-axis is the index of iterations. With $I_a$=1000, $I_b$=400, and $I_g$=1600, we split (by dotted lines) each generation to three parts: Alice pre-training, Bob pre-training, and interacting phase. The blue lines are generated by NIL with the pre-training of Bob while the red lines are generated when Bob is not pre-trained (here $I_g$=2000 to make a fair comparison). For the blue lines, $\mathbb{E}_{\mathcal{L}\sim P(\mathcal{L}|\mathrm{NIL})}[\rho(\mathcal{L})]$ will not change when Bob is pre-training (begins at the 1000th iteration), because Alice do not update parameters at that time. However, for the red lines, $\mathbb{E}_{\mathcal{L}\sim P(\mathcal{L}|\mathrm{NIL})}[\rho(\mathcal{L})]$ begin to decrease at the 1000th iteration. That is because when Bob is not pre-trained, the language learned by Alice may be impacted by playing with a fresh new Bob at the beginning of interacting phase! That is why the pre-training of Bob can make the NIL more efficient and robust. If Bob are pre-trained by the data generated by Alice in the current generation, Bob would be more "familiar" with Alice's language, and hence ensures a more stable interacting phase.

**Looking at the Emergent Languages**

From the discussions above, we know that NIL can ensure a high expected $\rho$ of the emergent language, and a high validation performance. Here we show the evolution of the distributions of emergent languages to provide a better intuition on how NIL works.

We first provide two examples of converged language (i.e., the language generated by Alice in the last generation) using the none-reset method and the resetting-both method in Table 6 and 7, respectively. In these examples, both languages can almost unambiguously represent all 64 different types of objects in $\mathcal{X}$, and hence they can help Alice and Bob to play the game successfully. However, the language generated using iterated learning has a clear compositional structure: the first position of the message represents different colors, and the second position represents the shape. Such a structure is quite similar to what humans do, e.g., combine an adjective and a noun to represent a complex concept.

To better illustrate the posterior probability of emergent languages as a function of the corresponding value of $\rho$ and the generation, we provide the 3D views of $P(\rho(\mathcal{L})|D_i, R_i)$ in 80 generations in Figure 12 and 13. The heat-map provided in Figure 11 can be considered as the top views of these 3D illustrations. In these two figures, the x-axis and y-axis represent the index of generation and the topological similarity, and the z-axis represents the probability of languages with a specific value of $\rho$, in a specific generation. To make the figures easier to read, we smooth the distribution of $\rho$ in each generation using linear interpolation (Boyd & Vandenberghe, 2004).

Figure 14-(a) and (b) compare the posterior distributions at some typical generations, which can also be considered as the side views of the 3D illustration from the direction of x-axis. In these figures, we find that the initial distribution of $\rho$ is not flat. That is because even the prior probability for each language is uniform, the amounts of languages with extremely high $\rho$ and low $\rho$ only occupy a small portion among all possible languages, as stated in (Brighton, 2002). Hence the initial probability of $\rho(\mathcal{L})$ is no longer uniform and has a bell shape which is similar to the Gaussian distribution. One new trend provided by these figures is that, in the none-reset case, the width of the curves in different generations do not change much, while in the resetting-both case, the width of the curves

will gradually decrease (i.e., becomes more peaky). Such a trends means when iterated learning is applied, language tend to converge to some high-$\rho$ types.

Figure 15-(a) and (b) track the ratio of languages with different values of $\rho$, which can also be considered as the side views of the 3D illustration from the direction of y-axis. In these figures, we divide all possible languages into five groups based on their topological similarity, i.e., languages with $\rho \leq 0.2$, $0.2 < \rho \leq 0.4$, $0.4 < \rho \leq 0.6$, $0.6 < \rho \leq 0.8$, and $0.8 < \rho$. We plot the ratio of these five different groups of languages at the end of each generation. From Figure 15-(a), we can see that the high-$\rho$ language, which is represented by the bold curve, always occupy a small portion. The topological similarity of the dominant languages are $\rho < 0.4$. However, in the resetting-both case, as illustrated in Figure 15-(b), the portion of high-$\rho$ language will increase significantly, which further verifies that the iterated learning can gradually make the high-$\rho$ language dominate in posterior.

|  | blue | green | cyan | brown | red | black | yellow | white |
|---|---|---|---|---|---|---|---|---|
| box | aa | fh | af | hh | cg | fc | ha | hf |
| circle | da | df | hb | db | fa | da | dh | fb |
| triangle | gc | ff | ge | gf | gg | fg | ge | he |
| square | ae | fb | be | bb | bg | fb | gb | ba |
| star | ad | fd | de | db | dg | fd | ce | hc |
| diamond | ac | dd | dc | db | dg | fd | dc | dd |
| pentagon | ad | fe | ef | bd | eg | fc | ee | ed |
| capsule | aa | dd | de | db | dg | gd | de | fh |

Table 6: Example of the converged language in none-reset case $\rho = 0.23$

|  | blue | green | cyan | brown | red | black | yellow | white |
|---|---|---|---|---|---|---|---|---|
| box | aa | ea | ba | ga | da | ca | ha | fa |
| circle | ab | eb | bb | gb | db | cb | hb | fb |
| triangle | ae | **eb** | be | ge | de | ce | he | fe |
| square | af | ef | bf | gf | df | cf | hf | ff |
| star | ac | ec | bc | gc | dc | cc | **dh** | fc |
| diamond | ad | ed | bd | gd | dd | cd | hd | fd |
| pentagon | ag | eg | bg | gg | dg | cg | hg | fg |
| capsule | ah | eh | bh | gh | **hc** | ch | hh | fh |

Table 7: Example of the converged language in resetting-both case $\rho = 0.93$

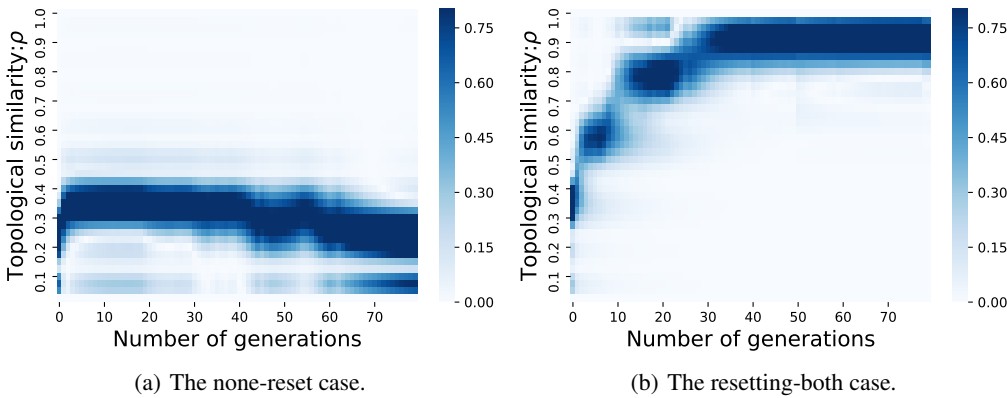

(a) The none-reset case.  (b) The resetting-both case.

Figure 11: Distribution of $P(\rho(\mathcal{L})|D_i, R_i)$ through 80 generations. Values of $\rho$ are divided into ten groups. The distribution of $\rho$ in each generation is smoothed using linear interpolation.

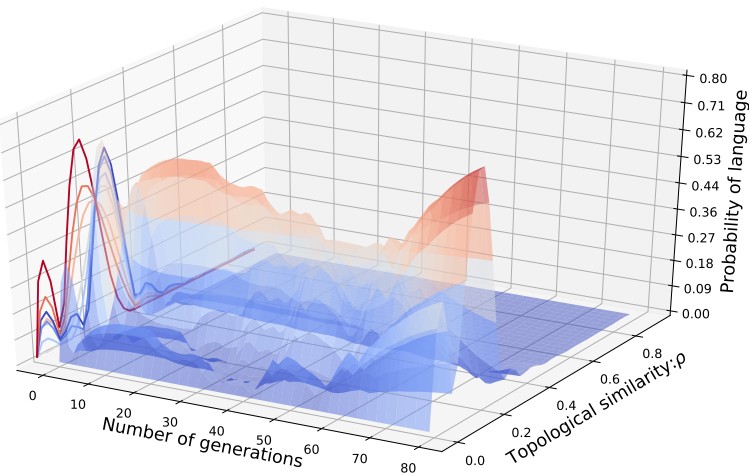

Figure 12: Language evolution of none-reset case in a 3D illustration.

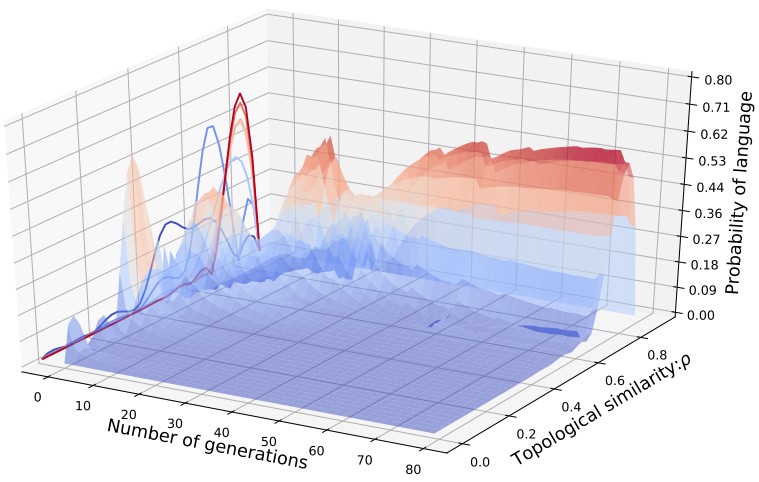

Figure 13: Language evolution of resetting-both case in a 3D illustration.

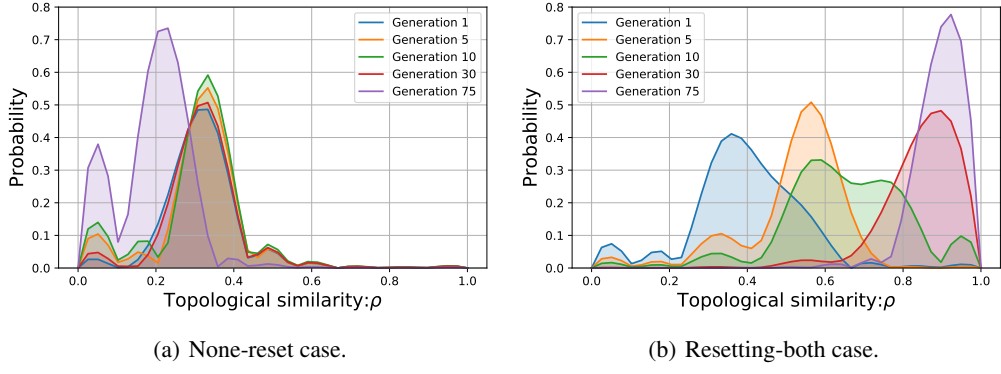

(a) None-reset case.

(b) Resetting-both case.

Figure 14: Distribution of $\rho(\mathcal{L})$ at different generations.

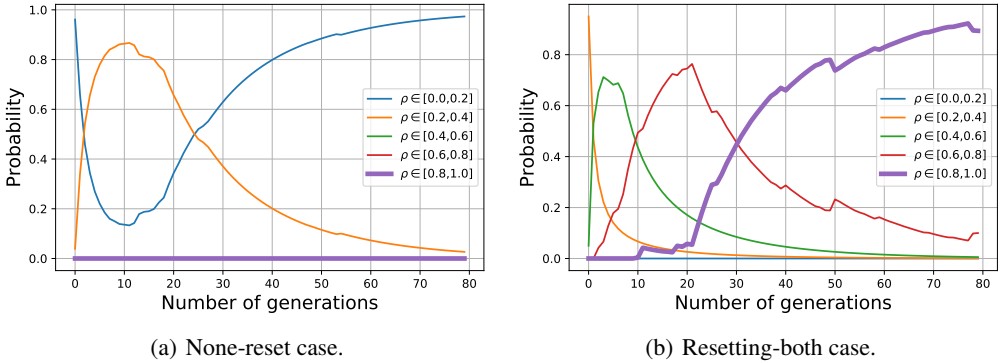

(a) None-reset case.

(b) Resetting-both case.

Figure 15: Evolution of language with different values of $\rho$.

