# OpenReview forum: "Compositional languages emerge in a neural iterated learning model"
_ICLR.cc/2020/Conference — Accept (Poster)_

### Official Review · AnonReviewer1 · 2019-10-23
**Official Blind Review #1**

**Rating:** 6

**Review:**

The paper "Compositional languages emerge in a neural iterated learning model" address the problem of language emergence in two-players games. In particular, the authors proposed a neural iterated learning model which seeks comopsitional languages. Authors claim that compositional languages are easier to be learned and that they allow listeners to more easily understand provided messages.

The problem of language emergence is interesting since it refers to the problem of finding efficient ways to communicate.  At first I was wondering why such compositional language messages would be desirable and was a bit negative on this work. But in Table 2 authors give results for zero-shot performance which emphasize the benefits resulting from finding such composition properties in language: compositional languages have greatly better generalization properties. I like the parallel that we can make with humans, that have to learn to understand language for achieving tasks when they are childs, which is simulated here in the reset and re-training performed  at the start of each generation, and which explains the natural emergence of such compositionality. This is not a big surprise, but I like the simple but clever idea of reset that the paper exploits.

What I like less is the inequality (5) that not fully convinced me. I am not sure why this should hold. Moreover, authors claim that if Ia is too long, no improvement of the topology can be made. I cannot understand why. For instance if there are ambiguous messages in D, the interaction phase can radically change the language even if the pre-training has converged... And why should weak pre-training favor low-p languages?

Also, the considered learning scheme is that in the transmitting phase Alice records messages for all objects in D. But is it realistic ?

A study of the impact of the size of vocabulary would also be useful (since it must have a big impact on the results)

At last, why considering such discrete messages while for agents communication  would be easier with continuous messages ? When considering continuous messages, the problem relates with disentanglement which is a current hot-topic: having one factor controling one specific aspect of the object would also be useful for improving zero-learning.

**Experience Assessment:**

I have read many papers in this area.

**Review Assessment: Checking Correctness Of Derivations And Theory:**

I carefully checked the derivations and theory.

**Review Assessment: Checking Correctness Of Experiments:**

I carefully checked the experiments.

**Review Assessment: Thoroughness In Paper Reading:**

I read the paper thoroughly.

---

> ### Author Response · Authors · 2019-11-11
> **General response to all comments**
>
> Thanks very much for the comments from all the reviewers. In the updated version, we mainly update the paper in the following aspects:
>
> 1. Condensed the main paper to 8 pages, and highlight our main contribution as “elaborated two main elements for the success of NIL, i.e. learning speed advantage and amplifying mechanism using probabilistic models, and use experimental results to verify them”.
>
> 2. Rewrote Appendix B to provide the necessary background when discussing different types of languages.
>
> 3. Added Appendix C to provide a formal definition of the agents involved in NIL, as well as the probability of any language and the expectation of $\rho$. We provide a thorough explanation of how our method can ensure the high-$\rho$ language gradually dominate in posterior probability, and how the degenerate component will be filtered during NIL.
>
> 4. Added Appendix D to articulate why hypotheses 1 and 2 can hold.
>
> 5. Extended Appendix E to discuss the robustness of our method by some experimental results on the corner cases. The influence of vocabulary size, message length, the role of Bob’s pre-training in its learning phase, are discussed.
>
> We hope the discussions added to those appendices can provide the reader with a better understanding of our method.

---

> ### Author Response · Authors · 2019-11-11
> **Replies to comments of Review # 1 (Part 1)**
>
> Many thanks to the reviewer for giving these insightful comments about the paper. The summary of the change of the paper can be found in “General response to all comments”.
>
> In general, we rewrite the probabilistic explanation part and try to make the paper easier to follow. Our replies to the questions are given as follows.
>
> 1：About compositionality and generalization of languages.
>
> [Summary of the issue]: The reviewer mentioned that “I like the parallel that we can make with humans, … This is not a big surprise, but I like the simple but clever idea of reset that the paper exploits.”
>
> [Our reply]: We thank the reviewer for pointing out the parallel of our work with how human learn and use natural languages. We cannot agree with the reviewer anymore.
>
> Actually, we believe the idea, “language is to use”, propose by [1], and believe that languages can be treated as feature representations of our experience and perceptions. As calculated in our paper, compositional languages occupy only a very tiny proportion of all possible languages. Thus, to us, it is very surprising that we humans could invent such sophisticated compositional language systems. We definitely agree that this amazing disentangling capability of human is a very key element to our intelligence, and it also should be a fundamental element to artificial intelligence.
>
> However, as our natural languages today are far so sophisticated, we believe that it should be much simpler and less complicated at their first emergences. Further, if we could find out the mechanism facilitating the emergence of such compositional languages/properties, it will be the very first step towards a more “real” artificial intelligence.
>
> [1] Wittgenstein, Ludwig. Philosophical investigations. John Wiley & Sons, 2009.
>
> 2. Concerns about how the learning advantage emerges,
>
> [Summary of the question]: The reviewer mentioned in paragraph 2 that “... the inequality (5) (now it is inequality (3)) that not fully convinced me...”
> [Our reply]: We apologize for not explaining how this inequality is derived. The detail of its deduction is provided in Appendix C (equation 11-13). In short, this inequality is not a strict result — it is our hypothesis and could be deduced from intuition and verified by experiments (we are also struggling to provide a rigorous proof).
>
> [Summary of the question]: The reviewer mentioned that “if Ia is too long, no improvement of the topology can be made. I cannot understand why”
> [Our reply]: This can also be explained by the framework provided in Appendix C and the discussion in section 4.3. In short, if Ia is too long, Alice[i] may perfectly learn Di, and hence has a same $\rho$ with Alice[i-1]. As the interacting phase has no preference for the high-$\rho$ language, the expected $\rho$ of Alice[i] would not be improved. Or in other words, the enhancement provided by pre-training Alice would be weaken when Ia is too large. From the results in Table 1, we find that larger Ia will make the $\mathbb{E}[\rho_{1: 10}]$ and $\mathbb{E}[\rho_{70: 79}]$ lower, which means the enhancement of $\rho$ becomes slower. However, as pre-training Bob can also motivates a higher $\rho$, their converged $\rho$ can also be high (to 0.9). For example, we find in the Ia=8000 case, the expected rho will reach 0.9 until the 500th generation (roughly).
>
> We think the influence of Ia can be better explained by Figure 2(c). In the figure, if Alice[i-1] feed Alice[i] with a language whose $\rho=0.28$ and set Ia=1000, then after pre-training, the $\rho$ of Alice would be greater than 0.42. Hence Alice[i] will feed Alice[i+1] a language whose $\rho$ is greater than 0.42. Thus the expected $\rho$ would increase fast. On the contrary, if we use Ia=8000, Alice[t] can only obtain a language whose $\rho=0.32$ (compared with 0.45 in the Ia=1000 case), which makes the $\rho$ increase slower. The worst case, if we use Ia=10, the expected $\rho$ would not increase.
>
> In summary, a too long Ia will hinder the enhancement of $\rho$ due to not taking advantage of the learning speed advantage at Alice side. As we mentioned in the paper, we can find the appropriate range of Ia by drawing the learning curves.

---

> ### Author Response · Authors · 2019-11-11
> **Replies to comments of Review # 1 (Part 2)**
>
>
> [Summary of the question]: The reviewer mentioned that “... For instance if there are ambiguous messages in D, the interaction phase can radically change the language even if the pre-training has converged...”
>
> [Our reply]: First of all, the ambiguous messages (we call it degenerated components) in D would be gradually filtered out in NIL. That is because that the REINFORCE method can almost perfectly filter those degenerate components during Bob’s pre-training and interacting phase. Remember, in REINFORCE, the parameters of agents only update when R=1, i.e. the message can unambiguously deliver information. We verified this by the following experiment. We record the number of different message types during NIL in Figure 8, and found that this metric keeps increasing during NIL and finally converged to roughly 60. As we only have 64 different objects (56 in the training set), the result means the degenerate components can be gradually filtered out during NIL. The case mentioned by the reviewer, i.e. D has many ambiguous component, might be very rare after several generations.
>
> Furthermore, even there may be some degenerate components in Di, and Alice learns it well, the interacting phase may not change the language “radically”. By directly observing the distribution of the output of two softmax layers of Alice, we find the probability of character selecting converges fast. The process of filtering degenerate component only alter a small portion of mappings. (We can add them to Appendix if necessary, but we think Figure 8 is enough to explain this problem.)
>
> [Summary of the question]: The reviewer mentioned that “And why should weak pre-training favor low-p languages”
>
> [Our reply]: We think that this is the most interesting part in this work. We use a toy example and several experiments to verify this in Appendix D. Figure 2-(c) is also a good example.
>
>
> 3. Concerns about feeding all objects to Alice to generate D
>
> [Summary of the question]: The reviewer thinks that feeding all objects to Alice when generating D is not realistic.
>
> [Our reply]: Actually, it is not necessary to do so when the object space becomes larger. The most important thing when generating D is to ensure each attributes are involved. For example, if the object space is 20*20=400 (20 colors and 20 shapes, i.e., O(N^2), sampling more than 2*(20+20)=80 (i.e., O(N)) can yield a good enough performance, based on our experiments. (We can add them to the appendix if necessary.) Or in practices, as all the information of the language is embedded in the parameters of Alice[i-1], we may design some other methods rather than sampling Di to transfer the knowledge. However, this is not the focus of this paper. The main point is that feeding all objects to generate D when message space is large is not necessary. The NIL has the potential to work on more complex problems.
>
> One more thing we find is that the size of D plays a similar role to the Ia, i.e. the extent of weak pre-training. Both of them can be considered as the “bottleneck” between agents in different generations. As the compositional languages can emerge from the pressure of expressivity and compressivity, using small Ia and small size of D can both provide this pressure. But due to the time constraint, we do not say too much about this in the paper. So in this paper, as the size of the object space is only 8*8=64, we feed all the objects to generate D and use Ia to explain the bottleneck effect. This may also make us to better understand the role of weak pre-training.
>
>
> 4. Influence of Vocabulary Size.
>
> [Our reply]: Thanks very much for this suggestion. We add the experimental results on different vocabulary size and message length in Table 5 of Appendix E. The algorithm is quite robust to the different valuesof these hyper-parameters.
>
>
> 5. Why only considering discrete messages.
>
> [Our reply]: Yeah, we are also working on how to combine NIL to disentanglement representation learning. However, it seems the discrete message plays an important role: we cannot even find the obvious learning speed advantage when the message is continuous. Anyway, we would try more in this direction.
>
> Another motivation for using discrete message for communication is to be close to the nature of human language. As our natural languages contain only discrete messages, we also follow this setting in our multi-agent systems. More importantly, if we take the multi-agent system as a simulation to the physical world, we could consider Bob as some listening human, or, more inspiringly, a NLP system. From this perspective, using discrete messages could be more inspiring and insightful for the NLU researches.
>
> Thanks again for the review and comments! Hope the above replies could help.

---

> ### Author Response · Authors · 2019-11-15
> **Any questions about our response?**
>
> Hi,
>
> We updated the response several days ago. As the deadline for rebuttal is approaching, we are wondering whether you have some concerns about our response. Or do you have any other comments on it? Thanks very much.

---

### Official Review · AnonReviewer3 · 2019-10-23
**Official Blind Review #3**

**Rating:** 6

**Review:**

This paper studies the emergence of compositional language in neural agents. They propose an iterated learning method that consists of three phases: a supervised learning phase for a randomly-initialized speaker and listener, a self-play phase (where both agents are updated together), and a phase where a new dataset is created based on the current speaker’s language. This dataset is then passed on to the next ‘generation’ of speaker and listener. The paper finds that this procedure, with the right hyperparameters, leads to the emergence of more compositional languages in a simple symbolic referential game.

The question of how to emerge a compositional language is indeed interesting. This paper does a good job of conducting a careful set of ablations and analyzing the results. In my mind, the main scientific contribution of this work is the empirical verification of the principle ‘compositional languages are easier to learn’. While this principle is intuitive, it’s good to see it confirmed via experiments. The paper’s description of the ‘interval of advantage’ --- the range of updates where a compositional language performs better on the task than a non-compositional language --- is insightful to me.

I do have concerns for this paper around utility and novelty. As the paper mentions, it has already been shown that iterated learning procedures give rise to more compositional languages in non-neural models. While there are some things to consider in adapting this to neural networks, to my eye they seem rather straightforward (i.e. tuning the number of updates of the speaker and listener, the values I_a and I_b), contrary to the paper’s assertion.  From a utility perspective, the paper doesn’t go into how this might be practically applied in general to train neural agents to learn compositional languages in more complex environments (where they might be simultaneously speakers and listeners), as they stick to a very simple symbolic referential game. The main contribution of this paper is really: “studying how neural networks behave when trained in an iterated learning setting in a simple referential game“. I think this is a nice contribution, but the main question for me is whether this is enough for an ICLR acceptance.

My other concern is around the length of the paper. In my opinion, while the paper is well-written, it’s quite bloated and there is a lot of repetition. I think the paper could easily be condensed to 8 pages and retain the same information. Alternatively, some of the graphs in the Appendix (which are quite nice) could be added to the main paper to give more insight about how neural networks behave in this iterated learning procedure.

Finally, the paper shows that compositional languages generalize better to the held-out validation set. While this is also an intuitive result, it’s nice to have in the paper. I’d encourage the authors to remove the ‘zero-shot’ terminology though (which usually refers to predictions on new samples outside of the training distribution), and just stick to what is actually being shown, which is improved generalization.

Overall, I like the paper, but due to the concerns mentioned above I think it’s borderline, with a slight lean towards rejection.


**Experience Assessment:**

I have published one or two papers in this area.

**Review Assessment: Checking Correctness Of Derivations And Theory:**

N/A

**Review Assessment: Checking Correctness Of Experiments:**

I assessed the sensibility of the experiments.

**Review Assessment: Thoroughness In Paper Reading:**

I read the paper at least twice and used my best judgement in assessing the paper.

---

> ### Author Response · Authors · 2019-11-11
> **General response to all comments**
>
> Thanks very much for the comments from all the reviewers. In the updated version, we mainly update the paper in the following aspects:
>
> 1. Condensed the main paper to 8 pages, and highlight our main contribution as “elaborated two main elements for the success of NIL, i.e. learning speed advantage and amplifying mechanism using probabilistic models, and use experimental results to verify them”.
>
> 2. Rewrote Appendix B to provide the necessary background when discussing different types of languages.
>
> 3. Added Appendix C to provide a formal definition of the agents involved in NIL, as well as the probability of any language and the expectation of $\rho$. We provide a thorough explanation of how our method can ensure the high-$\rho$ language gradually dominate in posterior probability, and how the degenerate component will be filtered during NIL.
>
> 4. Added Appendix D to articulate why hypotheses 1 and 2 can hold.
>
> 5. Extended Appendix E to discuss the robustness of our method by some experimental results on the corner cases. The influence of vocabulary size, message length, the role of Bob’s pre-training in its learning phase, are discussed.
>
> We hope the discussions added to those appendices can provide the reader with a better understanding of our method.

---

> ### Author Response · Authors · 2019-11-11
> **Replies to comments of Review #3 (Part 1)**
>
> Many thanks to the reviewer for giving these insightful comments about the paper. The summary of the change of the paper can be found in “General response to all comments”.
>
> In general, we rearrange the paper to highlight our contributions of exploring the mechanisms of NIL and the existence of learning speed advantage. We also articulate that applying iterated learning to neural agents is not straightforward, not as claimed by the reviewer. Our replies to the questions are given as follows.
>
> 1. Novelty.
>
> [Summary of the question]: The reviewer claims that the main contribution of this paper is “the empirical verification of the principle ‘compositional languages are easier to learn”, and “studying how neural networks behave when trained in an iterated learning setting in a simple referential game”.
>
> [Our reply]: We are sorry for not strengthening the difficulties for such an adaptation, and we do not agree with the reviewer on what is the main contribution of this paper.
>
> The iterated learning is proposed by a group of evolutionary linguists to explain the origin of compositionality in human language. They only use laboratory experiments to verify it. Hence the “non-neural models” mentioned by the reviewer is actually human. Even though the authors of [1] apply a Bayesian framework to explain it, the most important element of iterated learning, i.e. the preference of high-$\rho$ language, is not well explained, as the authors directly assign a higher prior  to the high-$\rho$ languages.
>
> Hence we think, verifying the fact that high-$\rho$ language is learned faster by both neural speaker and listener and explain it well are really good contributions to this field. We cannot directly say high-$\rho$ language is favored by neural agents because humans do so. Actually, the authors in [2] demonstrate that whether such a preference exists depends on the structure of the input signal. Due to the page limits, we do not discuss it in details in this paper.
>
> On the other hand, without specific design, simply applying the iterated fashion to the neural network may make the model easy to crash (our model crashed many times before we found the current design…). We provide many experiments in Appendix E to show that our design is robust. Meanwhile, a discussion on how these designs help are provided in Appendix C and D. To the best of our knowledge, although there are also some works of applying the idea of iterated learning in the field of emergent communication, our work can still provide many insights by our analysis. For example, [3] reset the listener and achieve a good result. But this paper do not answer why it could work, and the algorithm proposed by them can also be considered as the ‘Only-reset-Bob’ case in our paper (remember the both-reset case significantly outperforms the only-reset-Bob case). Authors of [4] use the clearing memory mechanism, but they do not find the learning speed advantage nor the bottleneck effect of the message passing process, which is essential to the success of iterated learning. The method proposed in [5] applies a different structure compared with ours, but lacks a clear discussion on the role played by the iterated learning.
>
> In summary, we believe that the reviewer may underestimate and misunderstand the contributions of this paper.  This may due to that we do not strengthen the fact that applying iterated learning to neural agents is not that straightforward, and every process in our design contribute to the final result. We rearrange the paper and highlight our contribution in this version. More importantly, we provide a probabilistic explanation of our NIL method, as well as the explanation on learning speed advantage which is essential for applying NIL to more complex tasks, which is missing in most of the related works. The concerns about the utility will be answered in the next response.
>
>
> [1] Kirby, Simon, et al. "Compression and communication in the cultural evolution of linguistic structure." Cognition 141 (2015): 87-102.
> [2] Shangmin Guo, Yi Ren, Sergii Gavrylov, Stella Frank, Ivan Titov and Kenny Smith, ”The Emergence of Compositional Languages for Numeric Concepts Through Iterated Learning in Neural Agents.” EmCom@NeurIPS 2019
> [3] Li, Fushan, and Michael Bowling. "Ease-of-Teaching and Language Structure from Emergent Communication." NeurIPS 2019.
> [4] Cogswell, Michael, et al. "Emergence of Compositional Language with Deep Generational Transmission." arXiv preprint arXiv:1904.09067 (2019).
> [5] Chaabouni, Rahma, et al. "Word-order biases in deep-agent emergent communication." arXiv preprint arXiv:1905.12330 (2019).

---

> > ### Comment · AnonReviewer3 · 2019-11-11
> > **Response to authors**
> >
> > I thank the authors for their detailed rebuttal. A point that I missed was that the prior machine learning work on using iterated learning directly assigned a higher prior probability to high-p languages. This raises the significance of the paper in my view.
> >
> > I appreciate that the paper shows robustness results in Appendix E, and overall I'm very impressed with the detailed results in all of the appendices. Above, the authors claim that 'simply applying the iterated fashion to the neural network may make the model easy to crash'. Unless I'm missing something, I don't see where these experiments where the model 'crashes' are documented in the Appendix, and I think this might be a useful thing to add.
> >
> > Overall though, I think one strength of this paper that I didn't fully appreciate was how extensive their experimental analysis was in the Appendix. Given the points above, and the fact that the paper was condensed to 8 pages, I feel comfortable raising my review to a 6.

---

> > > ### Author Response · Authors · 2019-11-12
> > > **Many thanks to reviewers for their quick response.**
> > >
> > > Thanks very much for the quick response from the reviewers. We updated quite a few things, so it may takes some time to go through them. We sincerely thank the reviewers for their quick reponse as well as their helpful suggestions.

---

> ### Author Response · Authors · 2019-11-11
> **Replies to comments of Review #3 (Part 2)**
>
>
> 2. Utility.
>
> [Summary of the question]: The reviewer thinks that the paper does not go into how the NIL might be practically applied in general to train neural agents to learn compositional languages in more complex environments. The reviewer claims that sticking to a very simple symbolic referential game limits the contribution of this paper.
>
> [Our reply]: We thanks the reviewer for pointing out the utility issue, but we do not agree with the claim that sticking to a simple model limits the potential of the method. On the contrary, we think providing a clear model and a reasonable explanation is the best way to demonstrate the utility of one method, rather than choosing a complex task but cannot explain why the method works. That is our main reason for choosing such a simple case: we can propose a reasonable explanation (even not riguous) to every details when applying NIL. The results also show that our design is quite robust (as claimed in Appendix E).
>
> Actually, before we stick to this simple case, we indeed conducted several experiments on the following complex cases. For example, when using the image input instead of symbolic input, the NIL can also performs significantly better ($\rho$>0.7) than the baseline method ($\rho$<0.3). However, as the images may not be perfectly mapped to the meaning space we want to study (e.g. the position, border, and the background may impact the agents, as discussed in [1]), we give up the image input and convert to the simplest symbolic setting.
>
> For the simultaneously speaking and listening agent case mentioned by the reviewer, we also explored this before we stick to the simplest case. Actually, we also have some experimental results on this setting (NIL works well in this case.). Our first setting follows the obverter principle provided in [2], i.e. letting the agents listen and speak alternatively and add addition training phase to ensure a symmetric language. However, we find that at any time instance, one agent plays the role of speaker and another plays the role of the listener, which is exactly the same with the simplest case we studied in this paper. The only difference introduced by this setting is the procedure of making the language symmetric, which is not directly connected to the NIL mechanism. We are quite sure that our method could work well in this setting.  More specifically, each agent has one speaker and one listener, we require the agents to speak to itself after the interacting phase to make the language symmetric. So, as the motivation of this paper is to articulate how NIL works, we do not use this complex setting in this paper. The reason for us not considering multiple agent is the same: the Alice-Bob model is the most fundamental case of NIL, if we can understand this case well, it could be easily extended to these complex settings.
>
> In summary, we do not think sticking to the simplest case harms the contribution of this paper. On the contrary, it enables us to provide a clear model and explanation of NIL, which can provide more insights to the research in this direction.
>
> [1] Lazaridou, Angeliki, et al. "Emergence of linguistic communication from referential games with symbolic and pixel input." ICLR(2018).
> [2] John Batali. Computational simulations of the emergence of grammar. Approaches to the evolution
> of language: Social and cognitive bases, 405:426, 1998
>
>
> 3. About the paper length
>
> [Our reply]: We thanks the reviewer for this suggestion. We have condensed the paper into 8 pages, and put all other details in the appendix.
>
> 4. About the terminology
>
> [Our reply]: We thanks the for pointing out this problem. We delete the ‘zero-shot’ term throughout the paper, and use the term “validation” instead.
>
> Thanks again for the review and comments! Hope the above replies could help.

---

> ### Author Response · Authors · 2019-11-13
> **Something in progress.**
>
> Regarding the “model crash case”, we are sorry that we do not put these experiments in the paper yet. Actually, at the beginning of the algorithm design, we find the algorithm usually converges to a low $\rho$ (roughly 0.4) under some random seeds, before we add the pre-training phase of Alice. We will add some more ablation studies to make this clear. Plus, we plan to add the experiments of the influence of $c$, i.e., the number of candidates, as well as the experiment results of using image input and the result of simultaneously speaking and listening. Due to the time limitations, we hope these will be ready for the camera ready version.

---

### Official Review · AnonReviewer2 · 2019-10-27
**Official Blind Review #2**

**Rating:** 6

**Review:**

This paper proposed a neural iterated learning algorithm to encourage the dominance of high compositional language in the multi-agent communication game. The author shows that the iterative training of two agents playing a referential game can incrementally increase the agent to use the language with high topological similarity. The authors also demonstrated that topological similarity is correlated with zero-shot performance. And Experiment results show the authors could propose alternative pre-training strategies for the neural agent can prefer high compositional language and achieve high task performance.

Emerging the compositional language from uniform prior can be very challenging, as mentioned in the paper, "high-\rho language only represents a small portion of all possible unambiguous language" and "high-\rho do not seem to be directly preferred during the interaction phase, they can be favored by the neural agent during the learning phase." I agree with the authors with respect to the difficulties of generating high-\rho language, but I have questions about the designed learning phrase, especially how to avoid the mode collapse during training.

With the first hypothesis, "high topological similarity improves the learning speed of the speaking neural agent", I agree that high topological language has less low sample complexity compared to random sampled low topological language. However, low topological language didn't necessarily lead to low sample complexity, for example, given a D consists of {a, a, a, a ...}, the sample complexity can be quite low and also with a high topological score. I wonder is the hypothesis still true in this case?

On the second hypothesis, a high-\rho language will be faster to success choosing the right object using fewer samples. I agree with the authors that compositional language can and will lead to better generalization ability. However, from Algorithm1, it seems Bob receives the message only update with its parameters. There is no change of language generation. I wonder how the update of Bob will help Alice to speak the more compositional language? More explicitly, to avoid mode collapse.

Exp 3 mainly tests the model with different \rho as the posterior probability. In Exp 4, I assume the posterior probability of the mapping is random (uniform), is that correct? It will be great if the confirm this since this is my major doubt when reading the paper.

As mentioned above, I understand that high topological similarity language both benefit from the speaker and listener. However, there are some corner cases that mode collapse will happen and It seems the model will hardly recover from that. From Table 3, it seems the authors have fixed vocabulary size 8, I'm wondering what happens with a large vocabulary size? will the model still learn to emerge the compositional language with a large vocab size?

**Experience Assessment:**

I have published one or two papers in this area.

**Review Assessment: Checking Correctness Of Derivations And Theory:**

N/A

**Review Assessment: Checking Correctness Of Experiments:**

I carefully checked the experiments.

**Review Assessment: Thoroughness In Paper Reading:**

I read the paper thoroughly.

---

> ### Author Response · Authors · 2019-11-06
> **Queries about the questions**
>
> Thanks very much for your reviewing and advices! They are really helpful.
>
> However, we're a little bit confused about some questions in the above review. To be specific,
>
> 1. You mentioned in paragraph 4 that '... from Algorithm 1, it seems Bob receives the message only update with its parameters. ...'. Which phase you are referring to? (learning phase or the interacting phase) Or should we carefully explain the role Bob played in BOTH phases?
>
> 2. You mentioned in paragraph 5 that 'Exp 3... Exp4...'. Are you referring to the results illustrated in Fig3-(b,c), or probably Fig 5?
>
> We think your answer would help us respond to your question more efficiently. Thanks very much!

---

> > ### Comment · AnonReviewer2 · 2019-11-06
> > **Clarification of the questions**
> >
> > 1. You mentioned in paragraph 4 that '... from Algorithm 1, it seems Bob receives the message only update with its parameters. ...'. Which phase you are referring to? (learning phase or the interacting phase) Or should we carefully explain the role Bob played in BOTH phases?
> >
> > - I mean learning the phrase. I algorithm 1, before the interactive phrase, "Bobi updates its parameters if rewarded"
> >
> > 2. You mentioned in paragraph 5 that 'Exp 3... Exp4...'. Are you referring to the results illustrated in Fig3-(b,c), or probably Fig 5?
> >
> > - Sorry for the confusion, Exp3 refer to Fig3 - b,c, and Exp4 means Fig4.
> >
> > In 4.1, the paper mentioned, " We record the game performance (i.e., the rate of successful object selections) and mean ρ of the object-message pairs exchanged by the neural agents every 20 rounds." I just want to confirm the initial posterior probability of the mapping is uniform, rather than designed mixed of high-\rho and low-\rho.

---

> > > ### Author Response · Authors · 2019-11-07
> > > **Thanks for the quick response**
> > >
> > > Thanks very much for your quick response.
> > >
> > > We're adding some experiment results to the appendix. Briefly speaking, we've experimented on the concerns about mode collapse and the method could overcome it, but we didn't add them into the paper due to the page limitation and that it's not the our main contribution. Currently, we're running more experiments to show stronger evidence that our IL method could overcome the mode collapse problem.
> > >
> > > Meanwhile, we're running other experiments to answer the other questions.
> > >
> > > Thanks.

---

> ### Author Response · Authors · 2019-11-11
> **General reponse to all comments**
>
> Thanks very much for the comments from all the reviewers. In the updated version, we mainly update the paper in the following aspects:
>
> 1. Condensed the main paper to 8 pages, and highlight our main contribution as “elaborated two main elements for the success of NIL, i.e. learning speed advantage and amplifying mechanism using probabilistic models, and use experimental results to verify them”.
>
> 2. Rewrote Appendix B to provide the necessary background when discussing different types of languages.
>
> 3. Added Appendix C to provide a formal definition of the agents involved in NIL, as well as the probability of any language and the expectation of $\rho$. We provide a thorough explanation of how our method can ensure the high-$\rho$ language gradually dominate in posterior probability, and how the degenerate component will be filtered during NIL.
>
> 4. Added Appendix D to articulate why hypotheses 1 and 2 can hold.
>
> 5. Extended Appendix E to discuss the robustness of our method by some experimental results on the corner cases. The influence of vocabulary size, message length, the role of Bob’s pre-training in its learning phase, are discussed.
>
> We hope the discussions added to those appendices can provide the reader with a better understanding of our method.

---

> ### Author Response · Authors · 2019-11-11
> **Replies to comments of Review #2 (Part 1)**
>
> Many thanks to the reviewer for giving these insightful comments about the paper. The summary of the change of the paper can be found in "General response to all comments".
>
> In general, we argue that the NIL is quite robust to the corner cases mentioned by the reviewer. Our replies to the questions are given as follows.
>
> 1. Influence of degenerate component:
>
> [Summary of the question]: The reviewer mentions in paragraph 2 that “... but I have questions about the designed learning phrase, especially how to avoid the mode collapse during training.”  The first thing the reviewer concerns is the influence of the degenerate components, i.e. “... for example, given D consists of {a, a, a, a, ...}” in paragraph 3. The reviewer wonders whether hypothesis 1 is still true in this case.
>
> [Our reply]: Hypothesis 1 is still true for such languages with many degenerate components. (We define a degenerate component as a mapping of several objects to the same message, which is further illustrated in Appendix B.) In fact, a pure degenerate language would be learned even faster than a compositional language.
>
> $But\ this\ fact\ does\ not\ affect\ the\ effectiveness\ of\ our\ algorithm$. That is because that the REINFORCE method can almost perfectly filter those degenerate components during Bob’s learning and interacting phase. Remember in REINFORCE, the parameters of agents only update when R=1, i.e. when the message can unambiguously deliver information. We conduct several experiments to verify our argument.
>
> First, we record the number of different message types during NIL in Figure 8. We see that this metric increases during NIL, and finally converged to roughly 60. As we only have 64 different objects (56 in the training set). The result means the degenerate components can be gradually filtered out during NIL. The case mentioned by the reviewer, i.e. many messages like {a, a, a, a, ...} in D, might be very rare after several generations.
>
> To further verify the robustness of our method when facing degenerate components, we design two challenging tasks for NIL:
> Degenerate initialized, in which we let Alice learn from a pure degenerate language at the beginning of each generation before it learns from Di;
> Degenerate mixed, on top of degenerate initialization, we mix the data pairs generated by a pure degenerate language to every Di and ensures the proportion of the degenerate pairs is more than 50%, which makes Alice easier to collapse to a degenerate language during the learning phase. (We think this task is unrealistically challenging because there is no reason to have such a noisy Di.)
>
> However, as illustrated in Figure 9 (a) and (b), the algorithm is very robust to these tasks: the NIL performs significantly better than the none-reset case, in terms of both $\rho$ and validation performance.

---

> ### Author Response · Authors · 2019-11-11
> **Replies to comments of Review #2 (Part 2)**
>
>
> 2. The role  of Bob in the learning phase:
>
> [Summary of the question]: The reviewer mentions in paragraph 4 that “... it seems Bob receives the message only update with its parameters... I wonder how the update of Bob will help Alice to speak the more compositional language...”.
>
> [Our reply]: To answer this question, we would like to explain the role played by Bob in NIL, especially in the learning phase. From the probabilistic explanation in Appendix C, it is obvious that the language is stored at Alice in each generation and Bob can only influence Alice in the interacting phase. Suppose we do not pre-train Bob in the learning phase. Then at the beginning of the interacting phase, Alice will play the referential game with a fresh new Bob, i.e. this Bob knows nothing about the language used by Alice. Hence Alice will gear its language towards the prior distribution when “teaching” Bob. In other words, the expected $\rho$ of Alice may be hindered by this “innocent” Bob. On the contrary, if Bob is pre-trained in the learning phase, Bob may be “more familiar with the language used by Alice”, and hence make the interacting phase more stable. Actually, as we mentioned in hypothesis 2, the pre-training of Bob may lead Bob to favor those languages with a higher-$\rho$ than Di. This preference may be imposed on Alice in the interacting phase after Bob is pre-trained.
>
> To verify our argument, we record the expected $\rho$ every 20 iterations in two generations and plot the results in Figure 10. In the figure, the pre-training of Alice covers 0-1000 time steps (i.e., Ia=1000). For the NIL with Bob pre-training, we use (Ib=400), hence 1000-1400 time steps is for Bob’s pre-training. Then the 1400-3000 time steps is the interacting phase. For the NIL without Bob pre-training, we let the agent directly step into the interacting phase to play the game, hence in the 1000-3000 times steps, Alice and Bob play the game and update the parameters together. It is clear that when Bob is not pre-trained, the $\rho$ of the language will be hindered by the innocent Bob: the value decreases after the 1000 time step. On the other hand,when Bob is pre-trained, the $\rho$ of language will first be stable (because Bob’s pre-training does not update Alice’s parameter), and then increase, because of the learning speed advantage at the listener side.
>
> In summary, the pre-training and re-initializing of Bob indeed makes the NIL more robust and efficient. In Figure 3(b) and 4(a), we see that the resetting-both case always performs better than only-Alice case. (The probability model provided in Appendix C can better explain this.)
>
>
> 3. Some concerns about Figure 3(b,c) (now it is Figure 2(a,b)) and Figure 4(b) (now it is Figure 3(b))
>
> [Our reply]: The experiments in Figure 2(a,b) aims to test the learning speed advantage of Alice or Bob. Hence we directly feed the languages with different $\rho$s (that are  recorded during NIL) to Alice and Bob separately. We can consider this process as ONE run of the pre-training with a given data set of specific $\rho$.
>
> Besides these two experiments (and also the challenging tasks in Figure 9), we do not interrupt the NIL algorithm. That is to say, the initial probability is uniform, we do not manipulate Di during NIL.
>
>
> 4. Influence of Vocabulary Size.
>
> [Our reply]: We thanks for the reviewer for this suggestion. We add the experimental results on different vocabulary size and message length in Table 5, Appendix E. The algorithm is quite robust to different values of these hyper-parameters.
>
> Thanks again for the review and comments! Hope the above replies could help.

---

> ### Author Response · Authors · 2019-11-15
> **Any questions about our response?**
>
> Hi,
>
> We updated the response several days ago. As the deadline for rebuttal is approaching, we are wondering whether you have some concerns about our response. Or do you have any other comments on it? Thanks very much.

---

### Decision · Program_Chairs · 2019-12-19

**Decision:**

Accept (Poster)

**Comment:**

This paper examines the correspondence between topological similarity of languages (correlation between the message space and object space) and ability to learn quickly in a situation of emergent communication between agents.

While this paper is not without issues, it does seem to present a nice contribution that all of the reviewers appreciated to some extent. I think it will spark further discussions in this area, and thus can recommend it for acceptance.